# Epidermal PAR-6 and PKC-3 are essential for larval development of *C. elegans* and organize non-centrosomal microtubules

**Victoria G Castiglioni[†], Helena R Pires[†], Rodrigo Rosas Bertolini, Amalia Riga, Jana Kerver, Mike Boxem\***

Division of Developmental Biology, Institute of Biodynamics and Biocomplexity, Department of Biology, Faculty of Science, Utrecht University, Utrecht, Netherlands

**Abstract** The cortical polarity regulators PAR-6, PKC-3, and PAR-3 are essential for the polarization of a broad variety of cell types in multicellular animals. In *C. elegans*, the roles of the PAR proteins in embryonic development have been extensively studied, yet little is known about their functions during larval development. Using inducible protein degradation, we show that PAR-6 and PKC-3, but not PAR-3, are essential for postembryonic development. PAR-6 and PKC-3 are required in the epidermal epithelium for animal growth, molting, and the proper pattern of seam-cell divisions. Finally, we uncovered a novel role for PAR-6 in organizing non-centrosomal microtubule arrays in the epidermis. PAR-6 was required for the localization of the microtubule organizer NOCA-1/Ninein, and defects in a *noca-1* mutant are highly similar to those caused by epidermal PAR-6 depletion. As NOCA-1 physically interacts with PAR-6, we propose that PAR-6 promotes non-centrosomal microtubule organization through localization of NOCA-1/Ninein.

**\*For correspondence:**
M.Boxem@uu.nl

[†]These authors contributed equally to this work

**Competing interests:** The authors declare that no competing interests exist.

## Introduction

Polarity is a near universal property of cells that is essential for establishing proper cellular architecture and function. Epithelial cells – one of the major polarized animal cell types – polarize along an apical–basal axis and establish molecularly and functionally distinct apical, basal, and lateral membrane domains. The boundary between apical and lateral domains is marked by the presence of cell–cell junctions that provide adhesion between cells and prevent unwanted paracellular passage of molecules. The polarization of epithelial cells is orchestrated by conserved cortical polarity regulators that establish opposing membrane domains through mutually antagonistic interactions. In metazoans, the partitioning defective (PAR) proteins Par3, Par6, and atypical protein kinase C (aPKC) play a central role in the establishment of epithelial cell polarity. These highly conserved polarity regulators are essential determinants of apical domain identity, and are required for the positioning, maturation, and maintenance of apical cell junctions (*Achilleos et al., 2010*; *Franz and Riechmann, 2010*; *Georgiou et al., 2008*; *Harris and Tepass, 2008*; *Harris and Peifer, 2005*; *Harris and Peifer, 2004*; *Hutterer et al., 2004*; *Izumi et al., 1998*; *Joberty et al., 2000*; *Leibfried et al., 2008*; *Lin et al., 2000*; *Totong et al., 2007*; *Wodarz et al., 2000*; *Yamanaka et al., 2001*).

Par6 and Par3 are both PDZ domain-containing scaffold proteins that can interact with each other, with aPKC, and with numerous other proteins. Par6 and aPKC form a stable subcomplex by interacting through their PB1 domains (*Hirano et al., 2005*; *Wilson et al., 2003*). The association of Par6–aPKC with Par3 is dynamic. In *C. elegans* zygotes, PAR-6/PKC-3 shuttle between a kinase inactive complex with PAR-3 that promotes anterior segregation, and an active complex with the small GTPase CDC-42 (*Aceto et al., 2006*; *Beers and Kemphues, 2006*; *Rodriguez et al., 2017*; *Wang et al., 2017*). In epithelia, Par3 can promote the apical recruitment of Par6–aPKC (*Franz and*

*Riechmann, 2010*; *Harris and Peifer, 2005*; *Hutterer et al., 2004*; *Joberty et al., 2000*; *Lin et al., 2000*; *Wodarz et al., 2000*). In mature epithelia, however, the bulk of Par3 segregates to the apical/lateral border, where it plays an essential role in the positioning and assembly of apical junctions (*Achilleos et al., 2010*; *Georgiou et al., 2008*; *Harris and Tepass, 2008*; *Harris and Peifer, 2004*; *Izumi et al., 1998*; *Leibfried et al., 2008*; *Totong et al., 2007*; *Yamanaka et al., 2001*). The segregation of Par3 from Par6–aPKC in epithelia depends on phosphorylation of Par3 by aPKC, and involves handoff of Par6–aPKC to Cdc42 and the epithelial-specific Crumbs polarity complex (*Bilder et al., 2003*; *Harris and Peifer, 2005*; *Hong et al., 2003*; *Krahn et al., 2010*; *Morais-de-Sá et al., 2010*; *Nagai-Tamai et al., 2002*; *Nunes de Almeida et al., 2019*; *Walther and Pichaud, 2010*).

In addition to interactions that mediate the subcellular localization of Par6–aPKC or Par3, both Par6 and Par3 can interact with effector proteins to connect cortical polarity with downstream pathways (*McCaffrey and Macara, 2009*). For example, Par3 modulates phospholipid levels by recruiting the lipid phosphatase PTEN to cell junctions (*Feng et al., 2008*; *Martin-Belmonte et al., 2007*; *Pinal et al., 2006*; *von Stein et al., 2005*), inhibits Rac activity by binding to and inactivating the RacGEF Tiam1 (*Chen and Macara, 2005*, p. 1; *Mertens et al., 2005*, p. 1), and mediates spindle positioning in *Drosophila* neuroblasts through recruitment of Inscuteable (*Schober et al., 1999*; *Wodarz et al., 1999*). For Par6, fewer downstream targets have been described. In mammals, Par6 can recruit the E3 ubiquitin ligase Smurf1 to promote degradation of the small GTPase RhoA, causing dissolution of tight junctions (*Ozdamar et al., 2005*; *Sánchez and Barnett, 2012*, p. 1; *Wang et al., 2003*). Par6 can also bind to the nucleotide exchange factor ECT2 to regulate epithelial polarization and control actin assembly at metaphase in dividing epithelial cells (*Liu et al., 2006*; *Liu et al., 2004*; *Rosa et al., 2015*). As high-throughput studies have identified multiple candidate binding partners that have not yet been investigated (*Boxem et al., 2008*; *Brajenovic et al., 2004*; *Giot et al., 2003*; *Grossmann et al., 2015*; *Hein et al., 2015*; *Huttlin et al., 2015*; *Lenfant et al., 2010*; *Luck et al., 2020*; *Waaijers et al., 2016*), additional interactions important for the functioning of Par6 and for linking cortical polarity to other processes involved in epithelial polarization likely remain to be discovered.

Despite the conserved requirements for Par6–aPKC and Par3 in epithelial cells there are important context and cell-type dependent differences in the functioning of these polarity proteins (*Pickett et al., 2019*; *St Johnston, 2018*). For example, in *Drosophila,* Bazooka (Par3) is not required for junction positioning or polarization of cells in the follicular epithelium (*Pickett et al., 2019*; *Shahab et al., 2015*), and in the adult *Drosophila* midgut, the canonical Par, Crumbs, and Scribble polarity modules are not essential for apical–basal polarity (*Chen et al., 2018*). In *C. elegans*, requirements for PAR-3 and PAR-6 in embryonic epithelia also vary. PAR-6 appears to be required for apical junction formation in all epithelia, including the epidermis, intestine, foregut, and pharyngeal arcade cells (*Montoyo-Rosario et al., 2020*; *Totong et al., 2007*; *Von Stetina et al., 2017*; *Von Stetina and Mango, 2015*). However, while arcade cells show a complete lack of polarization upon PAR-6 loss, foregut, intestinal, and epidermal epithelial cells still establish an apical domain (*Totong et al., 2007*; *Von Stetina and Mango, 2015*). PAR-3 is required for apical junction formation in embryonic intestinal and pharyngeal epithelia, but not in epidermal epithelial cells (*Achilleos et al., 2010*).

Studies of PAR-6, PKC-3, and PAR-3 in *C. elegans* have largely focused on embryonic tissues. Here, we make use of targeted protein degradation to investigate the role of PAR-6, PKC-3, and PAR-3 in larval epithelia of *C. elegans*. Ubiquitous depletion of PAR-6 and PKC-3, but not PAR-3, resulted in a larval growth arrest, demonstrating that these proteins are required for larval development. Through tissue-specific depletion, we identified an essential role for PAR-6 and PKC-3 in the *C. elegans* epidermis. Depletion in this tissue caused growth arrest, a failure to undergo molting, and severe defects in the division pattern of the epidermal seam cells. We also observed defects in the maintenance of apical cell junctions, and a failure to exclude LGL-1 from the apical domain. Finally, we identified a novel role for PAR-6 in organizing non-centrosomal microtubule arrays in the epidermis. Epidermal depletion of PAR-6 led to defects in the localization of the microtubule organizer NOCA-1/Ninein, as well as of the γ-tubulin ring complex component GIP-1, and of the sole Patronin/CAMSAP/Nezha homolog PTRN-1. Microtubule defects in a *noca-1* mutant closely resembled those in PAR-6 depleted animals, including the loss of GIP-1 localization. As NOCA-1 physically

interacts with PAR-6, we conclude that PAR-6 likely organizes non-centrosomal microtubule arrays through localization of NOCA-1.

## Results

### PAR-6 and PKC-3 are essential for larval development

To investigate the role of PAR-6, PKC-3, and PAR-3 in larval development, we made use of the auxin-inducible degradation (AID) system. The AID system enables targeted degradation of AID-degron tagged proteins through expression of the plant-derived auxin-dependent E3 ubiquitin ligase specificity factor TIR1 (*Nishimura et al., 2009*; *Zhang et al., 2015*; *Figure 1A*). Using CRISPR/Cas9, we inserted sequences encoding the AID-degron and the green fluorescent protein (GFP) into the endogenous *par-6*, *pkc-3*, and *par-3* loci, such that all known isoforms of each protein are tagged (*Figure 1B*). PAR-6 was tagged at the shared C-terminus, and PKC-3 at the N-terminus. The *par-3* locus encodes two groups of splice variants that use two alternative start sites, termed PAR-3L (for long) and PAR-3S (for short) (*Achilleos et al., 2010*; *Li et al., 2010a*). PAR-3L isoforms are expressed maternally and in larval stages, but not or at low levels in the embryo, while PAR-3S isoforms are expressed zygotically and in larval stages, but not maternally (*Achilleos et al., 2010*; *Li et al., 2010a*). To deplete both PAR-3L and PAR-3S isoforms, we inserted the GFP–AID tag at both start sites. To examine if the presence of the GFP–AID tags interfered with protein function, we examined the growth rates of the tagged strains. Homozygous animals were viable and showed the same growth rates as wild-type, indicating that the proteins are still functional (*Figure 1C–E*). Each protein was enriched at the apical membrane domain of epithelial tissues, including the pharynx, excretory canal, intestine and epidermis (*Figure 1F,G*). This matches previous observations in *C. elegans* larvae (*Li et al., 2010a*; *Li et al., 2010b*), and further indicates that the GFP–AID tag does not interfere with protein functioning. In the epidermis, we sometimes observed higher levels of fluorescence at the seam–seam junctions than at the seam–hyp7 junctions. Similar planar polarization of the PAR module was recently observed in the lateral epidermis of the elongating embryo (*Gillard et al., 2019*). However, in the larval epidermis we only observed planar enrichment in a subset of animals. Whether this pattern is functionally significant remains to be determined. Finally, to further investigate potential isoform-specific expression of PAR-3, we examined the expression of the PAR-3L isoforms alone during larval development (*Figure 1—figure supplement 1*). PAR-3L was expressed in the intestine, where it localized to the apical domain, but we observed little or no expression in the pharynx or epidermis (*Figure 1—figure supplement 1D*). Thus, PAR-3S appears to be the predominant isoform group in larval tissues.

To investigate the role of PAR-3, PAR-6 and PKC-3 in larval development we degraded each protein using a ubiquitously expressed TIR1 under the control of the *eft-3* promoter (*Zhang et al., 2015*). We tested the efficiency of protein degradation by exposing synchronized L3 larvae to auxin and examining protein expression. Apical enrichment of PAR-3, PAR-6, and PKC-3 became indistinguishable from background fluorescence within 1 hr of exposure to 4 mM auxin in the pharynx, excretory canal, intestine, and epidermis (*Figure 1G*). To examine if the depletion of PAR-6, PKC-3, or PAR-3 affected larval development, we degraded each protein by addition of auxin at hatching and measured animal growth rates. Ubiquitous degradation of PAR-3 did not cause a defect in larval growth, and animals developed into morphologically normal and fertile adults (*Figure 1E*). To determine if the lack of phenotype was due to an inherent technical problem with our approach, we also depleted PAR-3 in the germline and early embryos using *Pgld-1* driven TIR1. Addition of auxin to L4-stage animals resulted in 100% embryonic lethality in the next generation (n = 378), compared to 3.2% in non-auxin-treated controls (n = 591). Thus, the lack of a visible phenotype upon larval degradation indicates that the functions of PAR-3 are not essential for larval development. Alternatively, despite visual absence of GFP::AID::PAR-3, degradation may be incomplete, or animals may express unpredicted non-tagged larval-specific protein isoforms. In contrast to PAR-3, depletion of PAR-6 or PKC-3 from hatching caused a striking growth arrest with animals not developing beyond L1 size (*Figure 1C,D*). Thus, PAR-6 and PKC-3 are essential for early larval development, and we focused our further analysis on PAR-6 and PKC-3.

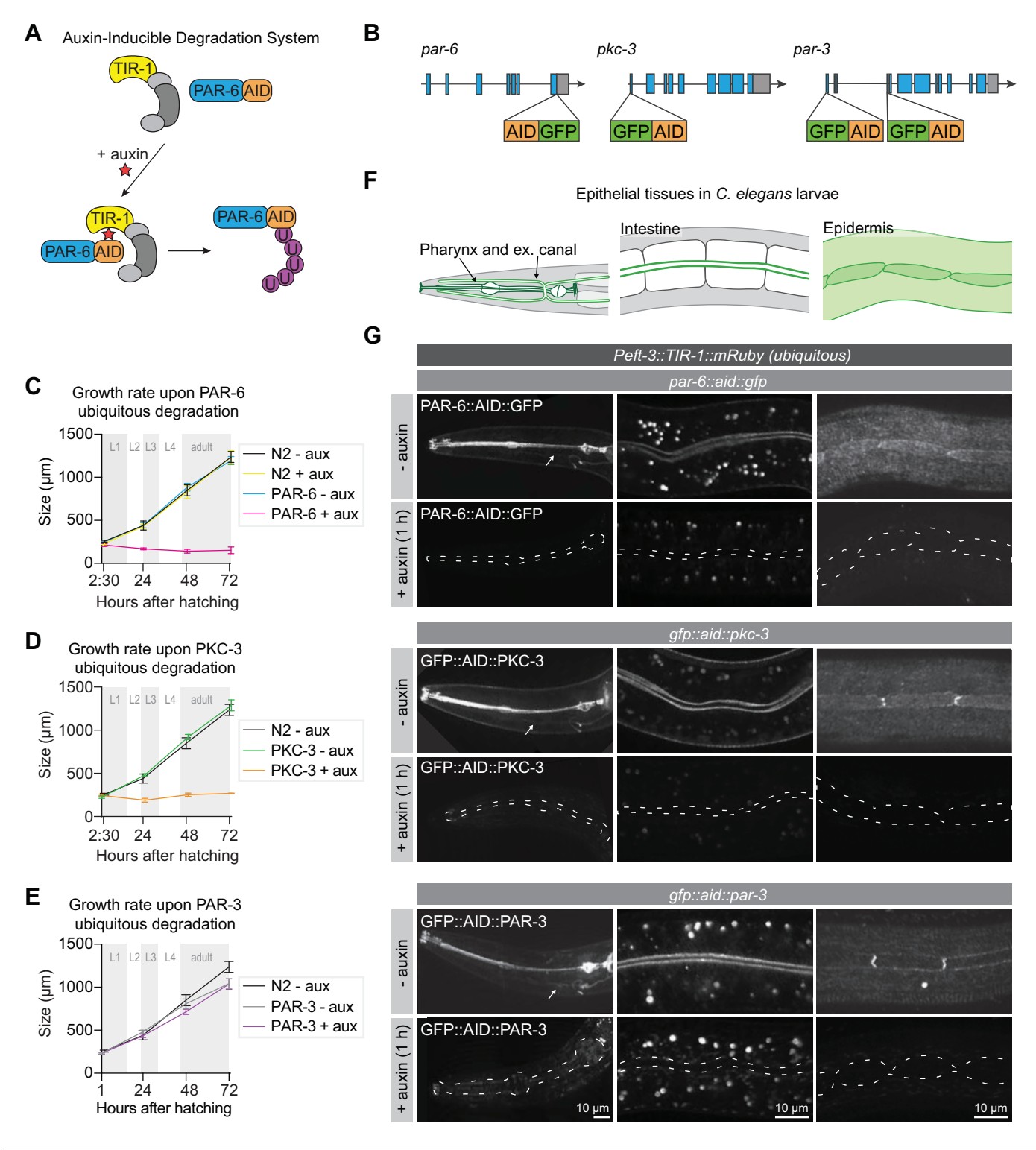

**Figure 1.** PAR-6 and PKC-3 are essential for larval development. (**A**) Overview of the AID system, which enables targeted degradation of AID-tagged proteins by the plant-derived E3 ubiquitin ligase specificity factor TIR1 upon addition of auxin. (**B**) Schematic representation of endogenous tagging of *par-6*, *pkc-3*, and *par-3* loci with sequences encoding a green fluorescent protein (GFP) and auxin-inducible degradation degron (AID) tag. (**C–E**) Growth curves of N2, *par-6::aid::gfp*, *gfp::aid::pkc-3*, and *gfp::aid::par-3* animals in absence (- aux) or presence (+ aux) of 4 mM auxin from hatching. Data show mean ± SD. Shading indicates the developmental stage of control animals. n = 6, 7, 8, and 8 for N2 - aux; 6, 7, 9, and 9 for N2 + aux; 7, 6, 9, and 9 for PAR-6 - aux; 8, 6, 7, and 9 for PAR-6 + aux; 22, 11, 10, and 14 for PKC-3 - aux; 19, 14, 9, and 10 for PKC-3 + aux; 10, 10, 10, and 10 for PAR-3 -

*Figure 1 continued on next page*

*Figure 1 continued*

aux, and 10, 10, 10, and 10 for PAR-3 + aux. (F) Graphical representation of larval epithelial tissues in *C. elegans*. Green indicates localization of PAR-6, PKC-3, and PAR-3. (G) Distribution of GFP::AID-tagged PAR-6, PKC-3, and PAR-3 in different larval tissues in absence (- auxin) or presence (+ auxin) of 4 mM auxin for 1 hr. Images are maximum intensity projections, and images of the pharynx are stitched montages. Dashed lines in - auxin panels outline pharynx (left panel), intestinal lumen (middle panel) or seam cells (right panel). White arrows point to the excretory canals.

The online version of this article includes the following source data and figure supplement(s) for figure 1:

**Source data 1.** Source data for *Figure 1*.

**Figure supplement 1.** Isoform-specific expression pattern of PAR-3.

**Figure supplement 1—source data 1.** Source data for *Figure 1—figure supplement 1*.

## PAR-6 and PKC-3 are essential in the larval epidermis, but not in the intestine

We next wanted to determine which larval tissue or tissues are severely affected by the loss of PAR-6/PKC-3 and contribute to the growth arrest. We focused on the two major epithelial organs: the intestine and the epidermis. The intestine is an epithelial tube formed in embryogenesis by 20 cells, which do not divide during larval development. PAR-6 and PKC-3 are highly enriched at the apical luminal domain (*Figure 2A*). The epidermis consists of two cell types: hypodermal cells and seam cells. The syncytial hypodermal cell hyp7 covers most of the body. Embedded within hyp7 are two lateral rows of epithelial seam cells, which contribute multiple nuclei to hyp7 through asymmetric divisions in each larval stage. PAR-6 and PKC-3 localize to the apical domain of the seam cells and hyp7 and are enriched at the seam–seam and seam–hyp7 junctions (*Figure 2B*).

To enable tissue-specific depletion of PAR-6 and PKC-3, we generated single-copy integrant lines expressing TIR1 in the intestine and epidermal lineages, using the tissue-specific promoters *Pelt-2* and *Pwrt-2*, respectively. In both tissues, protein depletion occurred within 1 hr of addition of 1 mM auxin (*Figure 2A–F*). To determine the contribution of the intestine and epidermis to the larval growth defects we observed above, we measured the growth rate of animals depleted of PAR-6 or PKC-3 from hatching in each tissue. Depletion of either protein from the intestine did not result in a growth delay or in obvious defects in morphology of the intestine (*Figure 2G,H*). These results are in contrast to the embryonic intestine, where PAR-6 has been shown to be required to maintain apical and junctional integrity (*Sallee et al., 2020*; *Totong et al., 2007*). Simultaneous depletion of PAR-6 and PKC-3 also did not result in a growth delay or visible abnormalities in the intestine (*Figure 2—figure supplement 1A*). These data indicate that PAR-6 and PKC-3 are not essential for the functioning and homeostasis of the larval intestine, though we cannot exclude that very low protein levels that we were not able to detect by fluorescence microscopy are sufficient in this tissue.

In contrast to the intestine, depletion of PAR-6 or PKC-3 from hatching in the epidermis caused an early larval growth arrest, as observed with ubiquitous degradation (*Figure 2I,J*). Thus, PAR-6 and PKC-3 play an essential role in the functioning and/or development of epidermal larval epithelia. We also noticed a small delay in growth in PAR-6::AID::GFP animals not exposed to auxin. One explanation is that TIR1 causes leaky degradation of PAR-6. However, no growth delay was observed in the absence of auxin when using ubiquitously expressed TIR1. Hence, the delayed growth may be due to other differences in genetic background. Finally, animals with ubiquitous PAR-6 or PKC-3 depletion have a more sick appearance than epidermal depleted animals, indicating that the functions of PAR-6 and PKC-3 are not limited to the epidermis. Indeed PAR-6 and PKC-3 were recently shown to be required for lumen formation in the excretory canal using a ZF degron-based protein degradation approach (*Abrams and Nance, 2020*).

## Cell autonomous and non-autonomous roles for PAR-6 and PKC-3 in growth, molting, seam-cell divisions, and seam-cell morphology

To characterize the growth arrest of PAR-6 and PKC-3 depleted animals in more detail, we examined arrested animals by Nomarski differential interference contrast (DIC) microscopy. We observed incompletely released cuticles 30 hr past exposure to auxin, indicative of molting defects (*Figure 3A*). To examine molting progression in more detail, we used a transcriptional reporter

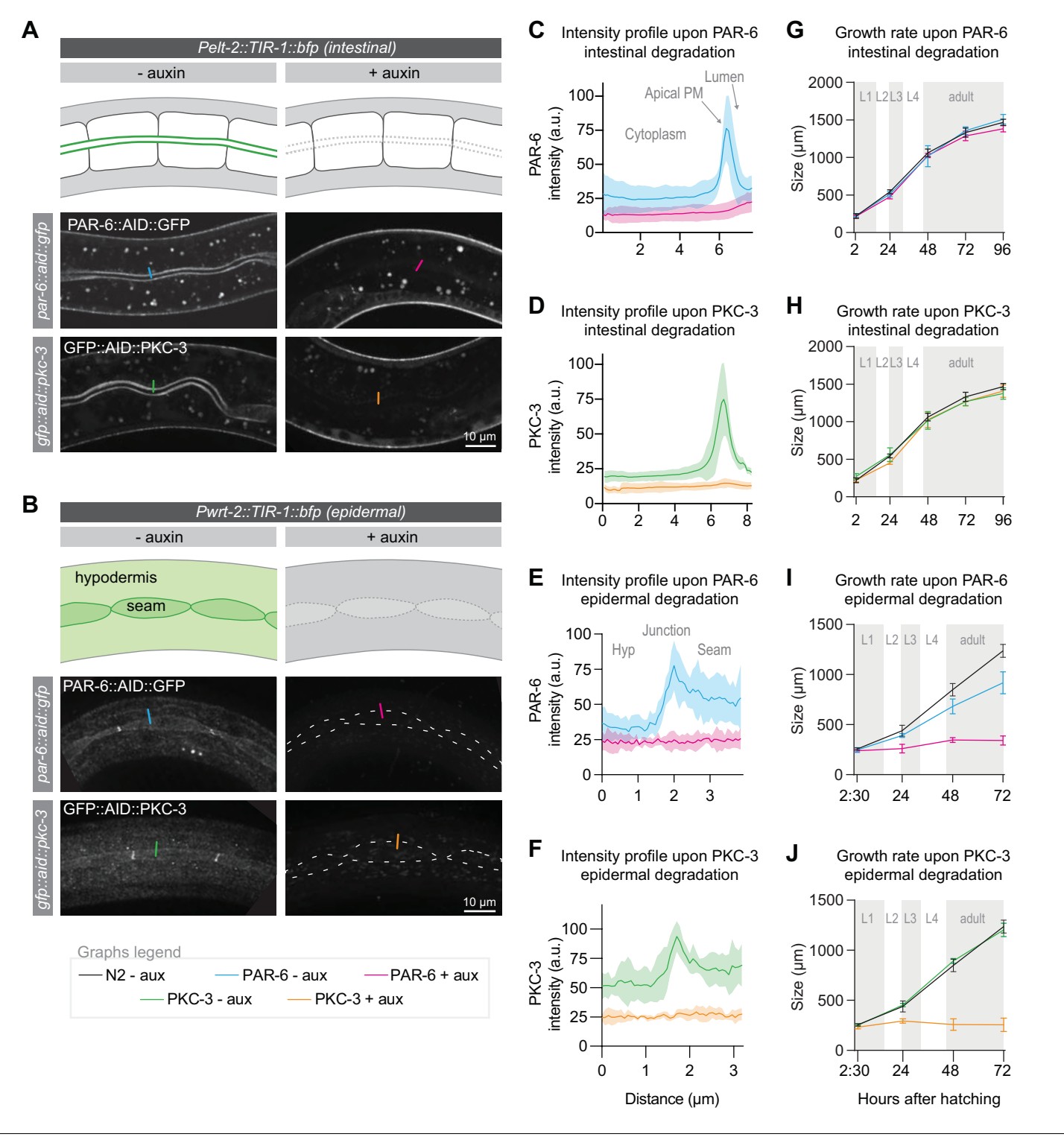

**Figure 2.** PAR-6 and PKC-3 are essential in the epidermis to support larval growth. (**A, B**) Distribution of PAR-6::AID::GFP and GFP::AID::PKC-3 in the intestine (**A**) and epidermis (**B**) in absence (- auxin) or presence (+ auxin) of 1 mM auxin for 1 hr. Images are maximum intensity projections of the luminal domain for the intestine, and the apical domain for the epidermis. Drawings are schematic representation of the area imaged, with the localization of PAR-6 and PKC-3 indicated in green shades. Gray indicates absence of PAR-6 and PKC-3. Short colored lines indicate the area quantified in C–F. (**C–F**) Quantification of apical GFP fluorescence intensity at the intestinal lumen and the hyp7–seam-cell junction in *par-6::aid::gfp* and *gfp::aid::pkc-3* animals in the absence (- aux) or presence (+ aux) of 1 mM auxin for 1 hr. Solid lines and shading represent mean ± SD. For the intestine, n = 10 animals for PAR-6 - aux, PAR-6 + aux, PKC-3 - aux, and PKC-3 + aux. For the epidermis, n = 8 animals for PAR-6 - aux, 6 for PAR-6 + aux, 5 for PKC-3 +

*Figure 2 continued on next page*

*Figure 2 continued*

aux, and 5 for PKC-3 - aux. (G–J) Growth curves of N2, *par-6::aid::gfp,* and *gfp::aid::pkc-3* animals in absence (- aux) or presence (+ aux) of 4 mM auxin from hatching. Solid lines and shading represent mean ± SD. In G and H, degradation was induced in the intestine, and in I and J in the epidermis. In the intestine, n = 13, 10, 13, 14, and 12 for N2 - aux; 7, 7, 7, 5, and 9 for PAR-6 - aux; 6, 6, 6, 5, and 7 for PAR-6 + aux; 8, 7, 8, 4, and 9 for PKC-3 - aux; and 8, 7, 8, 8, and 8 for PKC-3 + aux. In the epidermis, n = 6, 7, 8, and 8 for N2 - aux; 6, 5, 11, and 8 for PAR-6 - aux; 5, 10, 8, and 9 for PAR-6 + aux; 7, 7, 10, and 8 for PKC-3 - aux; and 8, 7, 12, and 13 for PKC-3 + aux.

The online version of this article includes the following source data and figure supplement(s) for figure 2:

**Source data 1.** Source data for *Figure 2*.

**Figure supplement 1.** PKC-3 does not act redundantly with PAR-6 in the *C. elegans* intestine.

**Figure supplement 1—source data 1.** Source data for *Figure 2—figure supplement 1*.

expressing GFP from the *mlt-10* promoter (*Meli et al., 2010*). *mlt-10* expression cycles, increasing during molting and decreasing during the inter-molt. Upon epidermal degradation of PAR-6 from hatching, *mlt-10* driven GFP levels remained low (*Figure 3B,C*), indicating that these animals fail to go through the L1/L2 molt.

To determine if the growth arrest and molting defects reflect a complete developmental arrest, we next examined the effects of PAR-6 depletion on the stereotypical division pattern of the seam cells. In every larval stage, an asymmetric cell division creates a new seam-cell daughter and a cell that differentiates to form neurons or fuse with hyp7 (*Chisholm and Hsiao, 2012a*; *Figure 3E*, blue shaded lineage tree). In the second larval stage, a symmetric division precedes the asymmetric division to double the number of seam cells. Depletion of PAR-6 from hatching did not disrupt the L1 asymmetric divisions, indicating that these animals are not developmentally arrested. As an additional marker of L1 development, we followed outgrowth of the excretory canals. During L1 development, both the anterior and posterior canals elongate from their initial positions at hatching to their final positions near the head and tail (*Figure 3—figure supplement 1A*; *Buechner et al., 2020*). Canal elongation still took place in PAR-6 depleted animals, with only a minor reduction in final length of the posterior canals (*Figure 3—figure supplement 1B*). Thus, PAR-6 depleted animals appear to continue the L1 developmental program, despite the lack of growth.

In contrast to L1 seam-cell divisions, the divisions that normally take place in the L2 stage were severely delayed (*Figure 3D,E*). At the time when control animals were already undergoing the L3 divisions, L2-stage divisions had still not taken place in PAR-6 depleted animals. Eventually, a next round of divisions did take place, but we observed various deviations from the normal L2 division pattern, including division failures and abnormal differentiation and fusion with hyp7. We did not observe any further divisions (*Figure 3E*). Following the delayed seam-cell divisions, we also observed numerous morphological abnormalities such as membrane protrusions, blebs, and abnormal division plane orientation (*Figure 3D*). Exposure of synchronized populations to auxin starting after the L1 or L2 divisions resulted in similar defects, indicating that seam-cell divisions require the functioning of PAR-6 throughout development (*Figure 3E*).

Expression of TIR1 under the *wrt-2* promoter results in degradation of target proteins in both the syncytial hypodermis and the seam cells. As the hypodermis is essential for molting and is involved in the control of larval growth (*Chisholm and Hsiao, 2012a*; *Chisholm and Xu, 2012b*; *Lažetić and Fay, 2017*), it is possible that the seam-cell defects are a secondary consequence of defects in the hypodermis. To address this, we expressed an exogenous copy of *par-6::mCherry* lacking the degron sequence using the hypodermal-specific *dpy-7* promoter (*Gilleard et al., 1997*). In combination with auxin-induced depletion of PAR-6::GFP::AID by *Pwrt-2* driven TIR-1, this results in absence of PAR-6 only from the seam cells. Hypodermal-specific expression of *par-6::mCherry* rescued the molting defects and seam-cell division delay observed upon PAR-6 epidermal degradation, and partially rescued the growth arrest (*Figure 3—figure supplement 1A–D*). However, seam-cell morphology defects and the abnormal cell division plane were not restored (*Figure 3—figure supplement 1C*).

To confirm that abnormalities in the hypodermis can affect seam-cell divisions, we used a CRISPR-tagged NEKL-2::AID strain that arrests growth and molting upon auxin addition (*Joseph et al., 2020*). Indeed, NEKL-2 depletion caused defects in the morphology of seam cells, as well as a partially penetrant reduction in seam-cell divisions, confirming that abnormalities in the hypodermis can affect the seam cells (*Figure 3—figure supplement 1E,F*).

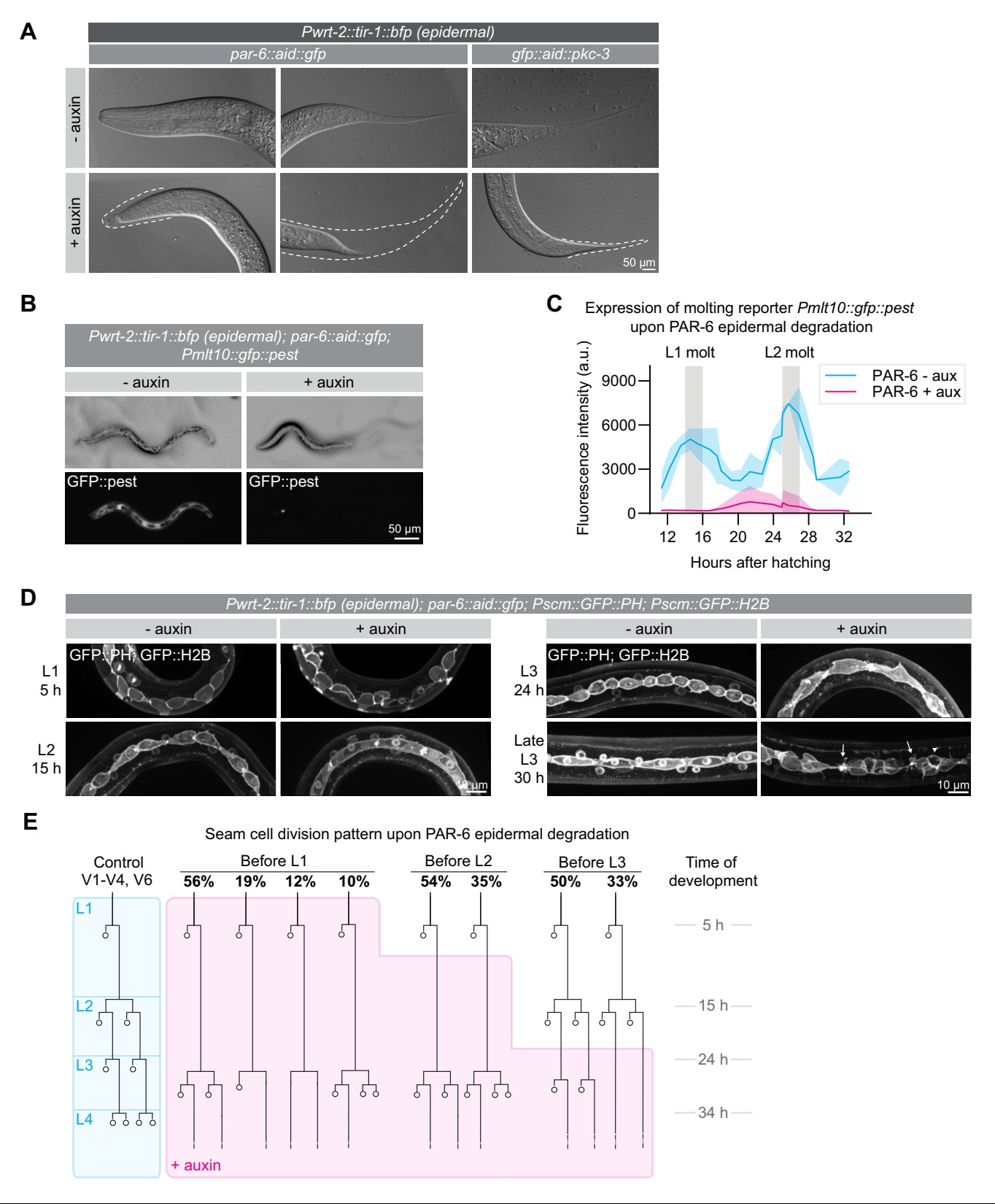

**Figure 3.** PAR-6 and PKC-3 are required in the epidermis for molting and seam-cell development. (**A**) DIC microscopy images of molting defects upon epidermal depletion of PAR-6 or PKC-3. Animals were grown in absence (-auxin) or presence (+auxin) of 1 mM auxin from hatching, and images were
*Figure 3 continued on next page*

*Figure 3 continued*

taken 30 hr after hatching. Dotted lines outline detached but unreleased cuticle in the pharynx and in the tail. Defects are observed in ~50% of the animals. Images are stitched montages. (B) Expression of the molting reporter *Pmlt-10::gfp::pest* in *par-6::aid::gfp* animals in the absence (- auxin) or presence (+ auxin) of 1 mM auxin from hatching. Images were taken at 22 hr of post-embryonic development. (C) Quantification of *Pmlt-10::gfp::pest* expression from 11 hr to 32 hr of post-embryonic development (mean fluorescence intensity ± SD) in *par-6::aid::gfp* animals in absence (- aux) or presence from hatching (+ aux) of 1 mM auxin from hatching. Measurements were done every hour. Each data point is an average of 3–12 measurements, with an average of 8 measurements per data point. (D) Examples of seam-cell division and morphology defects observed upon depletion of PAR-6::GFP::AID from hatching. Seam cells are visualized by expression of nuclear H2B::GFP and membrane-bound PH::GFP markers (*Wildwater et al., 2011*). Arrows indicate membrane protrusions and arrowhead indicates abnormal division plane orientation. Images are maximum intensity projections. (E) Seam-cell division pattern in *par-6::aid::gfp* animals in absence (control, blue) or presence (+ auxin, magenta) of 1 mM auxin. Auxin was added after hatching, before L2 divisions or before L3 divisions. For the control, n = 14, 75, 40, and 28 animals for the L1, L2, L3 and L4 divisions. For before L1, n = 17 animals for the L1 and 143 animals for the delayed L2 divisions. For before L2, n = 91 animals. For before L3, n = 40 animals.

The online version of this article includes the following source data and figure supplement(s) for figure 3:

**Source data 1.** Source data for *Figure 3*.
**Figure supplement 1.** Effect of PAR-6 epidermal degradation on canal outgrowth.
**Figure supplement 1—source data 1.** Source data for *Figure 3—figure supplement 1*.
**Figure supplement 2.** Hypodermal expression of PAR-6 is necessary for larval development.
**Figure supplement 2—source data 1.** Source data for *Figure 3—figure supplement 2*.

---

Taken together, our data show that PAR-6 is essential in the *C. elegans* hypodermis to support animal growth and molting. Whether the growth and molting phenotypes reflect separate functions of PAR-6, or are caused by the same underlying defect is difficult to establish, as molting is required for animal growth, but has also been reported to be governed by a size threshold (*Chisholm and Hsiao, 2012a*; *Lažetić and Fay, 2017*; *Uppaluri and Brangwynne, 2015*). The seam-cell division timing defects we observed appear to be a secondary consequence of hypodermal or molting defects. However, the fact that the growth arrest and seam abnormalities were not fully rescued by expression of PAR-6 in the hypodermis may indicate cell autonomous roles for PAR-6 in the seam, or alternatively that the *Pdpy-7::par-6::mCherry* rescue construct is not fully functional.

## PAR-6 and PKC-3 mediate apical LGL-1 exclusion and promote junction integrity in the larval epidermis

As one of the major functions of the apical PAR complex is to mediate the exclusion of basolateral proteins from the apical domain, we next examined the effects of PKC-3 depletion on two key aPKC target genes: LGL-1/Lgl and PAR-1. Both proteins are direct aPKC targets in epithelia, and in the *C. elegans* zygote their anterior exclusion is mediated by PKC-3 (*Beatty et al., 2010*; *Betschinger et al., 2003*; *Doerflinger et al., 2010*; *Hoege et al., 2010*; *Hurov et al., 2004*; *Motegi et al., 2011*; *Plant et al., 2003*; *Ramanujam et al., 2018*; *Yamanaka et al., 2003*). For these experiments we made use of integrated LGL-1::GFP transgene (*Waaijers et al., 2015*) and an endogenously tagged PAR-1::GFP fusion.

Depletion of PKC-3 in the intestine did not result in apical invasion of LGL-1 (*Figure 4—figure supplement 1A*). In contrast, degradation of PKC-3 in the epidermis resulted in clear apical LGL-1 localization in the seam cells within 6 hr of auxin addition (*Figure 4A,B*). Degradation of PKC-3 in the epidermis did not result in apical PAR-1 localization (*Figure 4C,D*). Instead, prolonged depletion of PKC-3 for 24 hr resulted in fragmentation of the normally contiguous PAR-1 signal at cell junctions, which may reflect an indirect effect of PKC-3 on junction organization (*Figure 4C*). These results demonstrate that PKC-3 is necessary to maintain the basolateral localization of LGL-1 in the seam cells, but not the intestine. In contrast, the apical exclusion of PAR-1 is not solely mediated by aPKC.

In embryonic epithelia, PAR-6 and PKC-3 are essential for proper junction formation, with loss of either protein resulting in fragmented cell junctions (*Montoyo-Rosario et al., 2020*; *Totong et al., 2007*). To investigate the requirement for PAR-6 and PKC-3 in junction integrity in larval epithelia, we assessed the localization of an endogenous mCherry fusion of the junctional protein DLG-1/Discs large upon degradation of PAR-6 or PKC-3 from hatching. In control animals not exposed to auxin,

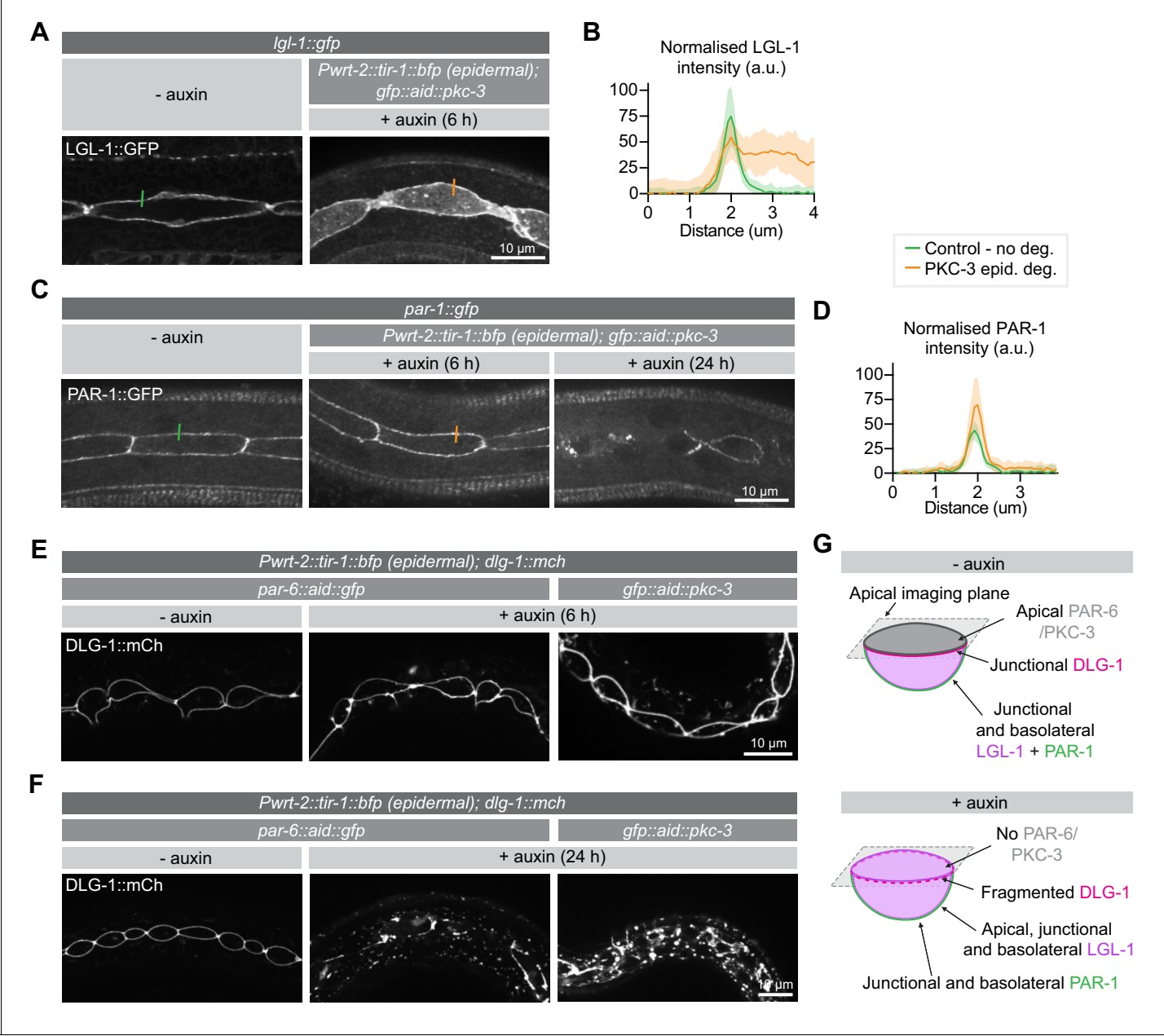

**Figure 4.** PKC-3 excludes LGL-1 from the apical cortex and, together with PAR-6, regulates junctions. (**A, B**) Distribution and quantification of LGL-1::GFP in the epidermis of *lgl-1::gfp* animals without auxin and in *lgl-1::gfp; gfp::aid::pkc-3; Pwrt-2::tir-1::bfp* animals in the presence of 4 mM auxin for 6 hr. Images are maximum intensity projections of the apical domain. Quantifications shows mean apical GFP fluorescence intensity ± SD at the hyp7–seam-cell junction, normalized to background intensity of each animal measured in the hypodermis. n = 7 animals for both conditions. Short colored lines in A indicate the area quantified in B. (**C, D**) Distribution and quantification of PAR-1::GFP in the epidermis in *par-1::gfp* animals without auxin and in *par-1::gfp; gfp::aid::pkc-3; Pwrt-2::tir-1::bfp* animals in the presence of 4 mM auxin for 6 or 24 hr. Images are maximum intensity projections of the apical domain. Quantifications show mean apical GFP fluorescence intensity ± SD at the hyp7–seam-cell junction, normalized to the background intensity of each animal measured in the hypodermis. n = 6 animals for both conditions. Short colored lines in C indicate the area quantified in D. (**E, F**) Junction organization visualized by DLG-1::mCherry expression in *par-6::aid::gfp* or *gfp::aid::pkc-3* animals in the absence (- auxin) or presence (+ auxin) of 1 mM auxin for 6 (**E**) or 24 (**F**) hours. Images are maximum intensity projections of the junctional domain. (**G**) Graphical representation of junctional defects in the seam cells upon PAR-6 or PKC-3 degradation.

The online version of this article includes the following source data and figure supplement(s) for figure 4:

**Source data 1.** Source data for *Figure 4*.

**Figure supplement 1.** PAR-6 and PKC-3 are not essential for LGL-1 localization or junction maintenance in the larval intestine.

**Figure supplement 1—source data 1.** Source data for *Figure 4—figure supplement 1*.

*Figure 4 continued on next page*

*Figure 4 continued*

**Figure supplement 2.** Localization dependencies of PAR-6 and, PKC-3.
**Figure supplement 2—source data 1.** Source data for *Figure 4—figure supplement 2*.

DLG-1 displays the typical ladder-like intestinal junction pattern and forms a continuous apical belt around the seam cells (*Figure 4—figure supplements 1B and E,F*). Upon degradation of PAR-6 in the intestine, we did not observe junctional defects (*Figure 4—figure supplement 1B*). We also did not observe any changes to the DLG-1 localization pattern in the epidermis after 6 hr of PAR-6 or PKC-3 depletion (*Figure 4E*). However, after 24 hr of degradation, DLG-1 no longer localized in a uniform band around the seam cells but appeared fragmented, with aggregates of bright DLG-1 interspersed with areas lacking fluorescent signal (*Figure 4F*). We also observed fluorescent aggregates in the hypodermis (*Figure 4F*). Thus, as in the embryo, PAR-6 and PKC-3 are essential for junction integrity in the epidermis. The fact that junctional defects took 24 hr to develop, compared to 6 hr for LGL-1 mislocalization, points to an inherent stability of cell junctions.

Finally, we investigated the localization dependencies between PAR-6 and PKC-3. Several studies demonstrated that PAR-6 and PKC-3 co-localize throughout development, and are mutually dependent on each other for their asymmetric localization (*Bossinger et al., 2001*; *Leung et al., 1999*; *McMahon et al., 2001*; *Nance et al., 2003*; *Nance and Priess, 2002*; *Tabuse et al., 1998*; *Totong et al., 2007*). Moreover, binding of PAR-6 to PKC-3 is required for apical localization of PAR-6, including in larval epithelia (*Li et al., 2010b*). Degradation of PAR-6 in the intestine resulted in the rapid loss of PKC-3 from the apical membrane domain, and degradation of PKC-3 similarly disrupted PAR-6 localization (*Figure 4—figure supplement 2A,B*). When we followed the apical loss of PKC-3 in the intestine over time, we observed similar dynamics of PAR-6 depletion and PKC-3 loss (*Video 1*). In the epidermis, the levels of PAR-6 and PKC-3 are more difficult to determine accurately, due to the low levels of expression of these proteins and the aggregation due to the mCherry reporters used. Nevertheless, depletion of PAR-6 resulted in a loss of the junctional enrichment of PKC-3, and vice versa (*Figure 4—figure supplement 2C–D*). These disruptions occurred rapidly, within 1 hr of auxin addition. Our results thus confirm the interdependency between PAR-6 and PKC-3.

## PAR-6 and PKC-3 control the organization of non-centrosomal microtubule arrays in the hypodermis

The loss of PAR-6 or PKC-3 affected several epidermal processes in which cytoskeletal elements play important roles, including molting, seam-cell divisions, and maintaining proper seam-cell morphology. The PAR proteins play essential roles in organizing the actomyosin cytoskeleton and microtubules in different settings, including asymmetric cell division, neuronal differentiation, and epithelial polarization (*Goldstein and Macara, 2007*; *Rodriguez-Boulan and Macara, 2014*; *St Johnston, 2018*). We therefore investigated if PAR-6 degradation affects the organization of actin or microtubules in the epidermis. To assess the organization of the actin cytoskeleton we used an epidermal transgene expressing the actin-binding-domain of *vab-10* fused to *mCherry* (*Gally et al., 2009*). We depleted PAR-6 from hatching and examined actin organization after 24 hr, when control larvae are in late L2 stage. Consistent with previous observations (*Costa et al., 1997*), we observed prominent circumferential actin bundles in hyp7, strong enrichment of actin along the hyp7–seam junctions,

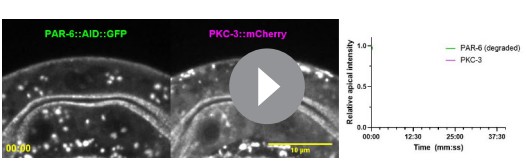

**Video 1.** Time-lapse imaging of PAR-6::AID::GFP and PKC-3::mCherry in animals expressing intestine-specific TIR1 upon addition of 1 mM Auxin.
https://elifesciences.org/articles/62067#video1

and largely anterior/posteriorly organized actin within the seam cells of control animals at this stage (*Figure 5A*). Upon PAR-6 depletion, actin organization appeared largely undisturbed in both the seam cells and hypodermis (*Figure 5A*), and actin bundles in hyp7 remained perpendicular to the seam cells (*Figure 5B*). These data indicate that PAR-6 does not play a major role in regulating the actin cytoskeleton in the *C. elegans* larval epidermis.

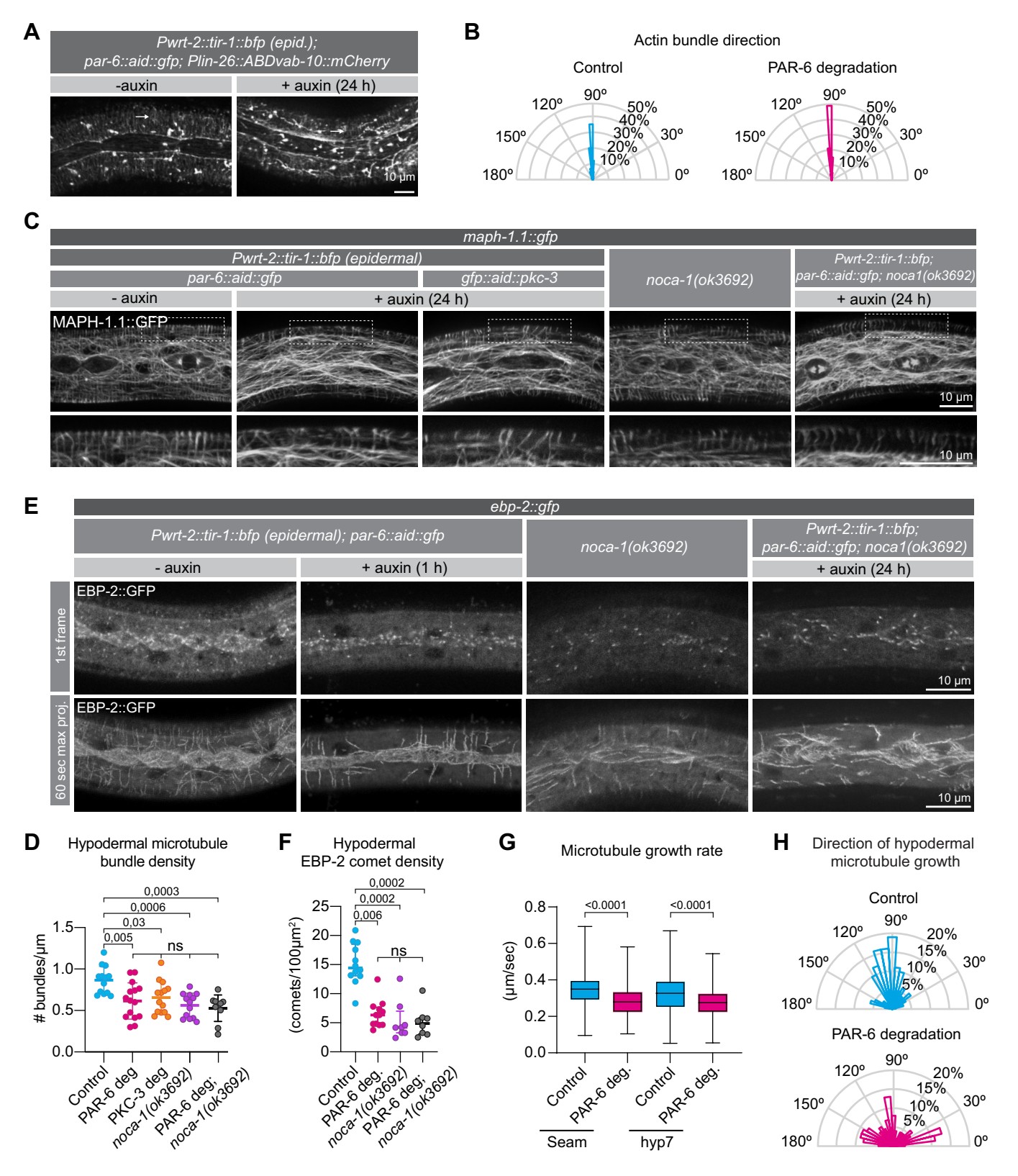

**Figure 5.** PAR-6 and PKC-3 control microtubule organization in the *C.elegans* epidermis. (**A**) Actin organization visualized by the *Plin-26::ABDvab-10:: mCherry* reporter in *par-6::aid::gfp* animals in absence (- auxin) or presence (+ auxin) of 1 mM auxin for 24 hr. Images are maximum intensity

*Figure 5 continued on next page*

Figure 5 continued

projections. (B) Quantification of actin bundle orientation. Angle is measured relative to the anterior (180°) – posterior (0°) axis. n = 100 bundles in five animals per condition. (C) Microtubule organization of the indicated genotypes visualized by MAPH-1.1::GFP in absence (- auxin) or presence (+ auxin) of 1 mM auxin for 24 hr. Images are maximum intensity projections. (D) Hypodermal microtubule bundle density. n = 13 animals for control, 15 for PAR-6 deg, 14 for PKC-3 deg, 13 for *noca-1(ok3692)*, and 10 for PAR-6 deg. in *noca-1(ok3692)*. Bars show mean ± SD. (E) Microtubule growth visualized by the plus end marker EBP-2::GFP in absence (- auxin) or presence (+ auxin) of 1 mM auxin for 1 hr. Images are a single frame or a 60 s maximum projection (one frame/second). To match the age of animals in (C), we depleted PAR-6 for 1 hr starting with 23 hr old L2 animals. (F) EBP-2 comet density in hyp7 in 24 hr old animals. n = 12 animals for control and PAR-6 deg, 8 for *noca-1(ok3692)*, and 10 for PAR-6 deg in *noca-1(ok3692)*. Auxin was present for 1 hr, from 23 to 24 hr of development. Bars show mean ± SD (G) Microtubule growth rate in 24 hr old animals. n > 400 comets in two animals (seam), four animals (hyp7 control), or five animals (hyp7 + auxin). Auxin was present for 1 hr, from 23 to 24 hr of development. Bars = mean ± SD (H) Quantification of microtubule growth orientation in hyp7 in 24 hr old animals. Auxin was present for 1 hr, from 23 to 24 hr of development. Vertical axis: left/right orientation; horizontal axis: anterior/posterior orientation. n = 150 comets. Bars = mean ± SD. Tests of significance: Tukey's multiple comparisons test for D, and Dunn's multiple comparisons test for F and G. ns = not significant.

The online version of this article includes the following source data for figure 5:

**Source data 1.** Source data for *Figure 5*.

We next inspected the organization of the microtubule cytoskeleton using an endogenously GFP tagged variant of the microtubule-binding protein MAPH-1.1 (*Waaijers et al., 2016*). We degraded PAR-6 in the epidermis from hatching and assessed the organization of epidermal microtubule arrays. In control animals, we observed highly ordered circumferential microtubule bundles in the dorsal and ventral sections of hyp7 underlying the muscle quadrants, and a microtubule meshwork in the lateral sections of hyp7 abutting the seam cells, as previously reported (*Chuang et al., 2016*; *Costa et al., 1997*; *Taffoni et al., 2020*; *Wang et al., 2015*; *Figure 5C*). In the seam cells the microtubule network was less well defined but also forms a meshwork (*Figure 5C*). In PAR-6 depleted animals, after 24 hr of development we observed a significant reduction in the density of circumferential microtubule bundles in the hypodermis (*Figure 5C,D*). Epidermal depletion of PKC-3 resulted in similar defects (*Figure 5C,D*). To understand the cause of the reduced microtubule density, we investigated microtubule dynamics using an endogenous fusion of the microtubule plus-end tracking protein EBP-2[EB1] to GFP (*Videos 2* and *3*). In control animals, EB1 comets moved predominantly in a circumferential direction, consistent with the organization of microtubule bundles in the epidermis, and both comet density and growth rates matched previous reports (*Figure 5E–H*; *Chuang et al., 2016*; *Taffoni et al., 2020*; *Wang et al., 2015*). Already within 1 hr of inducing depletion of PAR-6, we observed reduced microtubule dynamics (*Figure 5E–G*). The density of growing MTs was reduced by 56% (*Figure 5F*), and microtubule growth rate was reduced by 14% in hyp7 and by 16% in the seam cells (*Figure 5G*). These results suggest that the reduced density of microtubule bundles upon depletion of PAR-6 is the result of reduced growth of microtubules. We

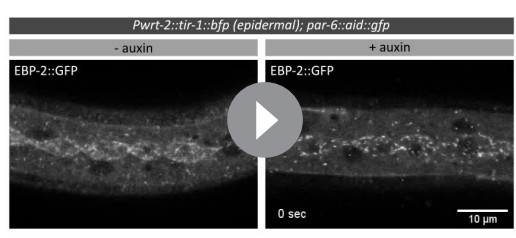

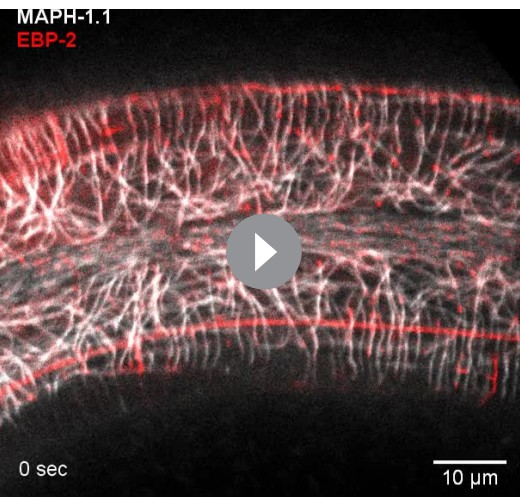

**Video 2.** Time-lapse imaging of EBP-2::GFP in control animals and PAR-6 depleted animals. Freeze frame circles EBP-2 comets as an example of quantification. Final frames show a time projection as displayed in *Figure 5E*.
https://elifesciences.org/articles/62067#video2

**Video 3.** Dual-color time-lapse imaging of EBP-2:: mKate2 and MAPH-1.1::GFP. New microtubules largely grow along existing bundles.
https://elifesciences.org/articles/62067#video3

also observed a defect in the directionality of microtubule growth. While 54% of the comets in control animals travel perpendicular to the seam cells (70–110°), this number is reduced to 24% upon PAR-6 degradation (*Figure 5H*), consistent with the defects in organization observed with GFP::MAPH-1.1.

## PAR-6 controls microtubule organization through its interaction partner NOCA-1/Ninein and the γ-tubulin ring complex

Two large-scale protein–protein interaction mapping studies in *C. elegans* had identified the microtubule-organizing protein NOCA-1 as an interaction partner of PAR-6 (*Boxem et al., 2008*; *Lenfant et al., 2010*). Affinity purification experiments showed that PAR-6 interacts with NOCA-1 through its PDZ domain (*Lenfant et al., 2010*), and we were able to confirm the PAR-6 PDZ interaction with NOCA-1 by yeast two-hybrid (*Figure 6—figure supplement 1*). NOCA-1 functions together with γ-tubulin to assemble non-centrosomal microtubule arrays in multiple tissues, including the epidermis, and is thought to be a functional homolog of the vertebrate microtubule organizer Ninein (*Green et al., 2011*; *Wang et al., 2015*). NOCA-1 localizes to the apical cortex in seam cells, similar to the localization of PAR-6 (*Figures 1G* and *6A*), but the mechanisms that mediate apical localization of NOCA-1 are currently not known. The physical interaction between PAR-6 and NOCA-1 prompted us to investigate if PAR-6 regulates non-centrosomal microtubule arrays through NOCA-1.

We first examined the effect of epidermal PAR-6 depletion on the localization of NOCA-1. To visualize NOCA-1 we made use of an existing transgenic line that expresses the epidermal-specific NOCA-1d and e isoforms fused to GFP from their endogenous promoter (*Wang et al., 2015*). In untreated control animals, we observed punctate localization of NOCA-1 in the epidermis, mostly clustered at the seam–seam and seam–hyp7 junctions, as previously observed (*Figure 6A*; *Wang et al., 2015*). Addition of auxin to induce epidermal PAR-6 degradation led to a 61% reduction in junctional levels of NOCA-1 within 6 hr (*Figure 6A,B*). Depletion of PKC-3 resulted in a similar reduction in NOCA-1, (*Figure 6G,H*). These results demonstrate that PAR-6 and PKC-3 promote the apical localization of NOCA-1. Because of the physical interaction between PAR-6 and NOCA-1, we hypothesize that the loss of PKC-3 indirectly affects NOCA-1 through loss of PAR-6 localization.

NOCA-1 was reported to work together with γ-tubulin and redundantly with Patronin/PTRN-1 in controlling circumferential microtubule bundle organization in the hypodermis (*Wang et al., 2015*). We therefore examined the effect of PAR-6 depletion on the localization of PTRN-1 and GIP-1, a core component of the γ-tubulin ring complex (γ-TuRC) required to localize other γ-TuRC components to the apical non-centrosomal microtubule-organizing center (ncMTOC) in the embryonic intestine (*Sallee et al., 2018*). To visualize PTRN-1 and GIP-1 we used endogenous PTRN-1::GFP and RFP::GIP-1 fusion proteins. GIP-1 localized in a punctate pattern at the seam–seam and seam–hyp7 junctions, similar to NOCA-1 (*Figure 6C*; *Sallee et al., 2018*; *Wang et al., 2015*). PTRN-1 also localized in a punctate pattern, but dispersed through the epidermis and lacking the junctional enrichment seen for NOCA-1 and GIP-1 (*Figure 6E*; *Wang et al., 2015*). Upon PAR-6 degradation, junctional GIP-1 levels were strongly reduced (*Figure 6C,D*), similarly to NOCA-1. We also observed that PAR-6 depletion resulted in a decrease in the number of PTRN-1 puncta in the epidermis (*Figure 6E,F*). As NOCA-1 is a direct interaction partner of PAR-6, we examined if the loss of GIP-1 is due to the loss of NOCA-1 localization. Indeed, in a *noca-1(ok3692)* deletion mutation GIP-1 levels were significantly reduced (*Figure 6I,J*), suggesting that NOCA-1 acts upstream of GIP-1 in the *C. elegans* larval epidermis.

Finally, we examined if the failure to properly localize NOCA-1 could explain the microtubule defects we observed upon PAR-6 depletion. We determined microtubule bundle density, EB1 comet density, and microtubule growth rate in *noca-1(ok3692)* animals. In the *noca-1(ok3692)* deletion mutant, we observed a significant reduction in the density of circumferential microtubule bundles in the hypodermis (*Figure 5C,D*). As reported in a previous study (*Wang et al., 2015*), we also observed reduced microtubule dynamics, with the density of growing MTs reduced by 65%, and microtubule growth rates reduced by 65% (*Figure 5E,G*). These values are all very similar to those we observed upon PAR-6 depletion. To further determine if *par-6* and *noca-1* act in a linear pathway, we degraded PAR-6 in *noca-1(ok3692)* mutant animals. Both microtubule bundle density and microtubule dynamics were reduced to a similar extent as in PAR-6 depleted or *noca-1* mutant animals

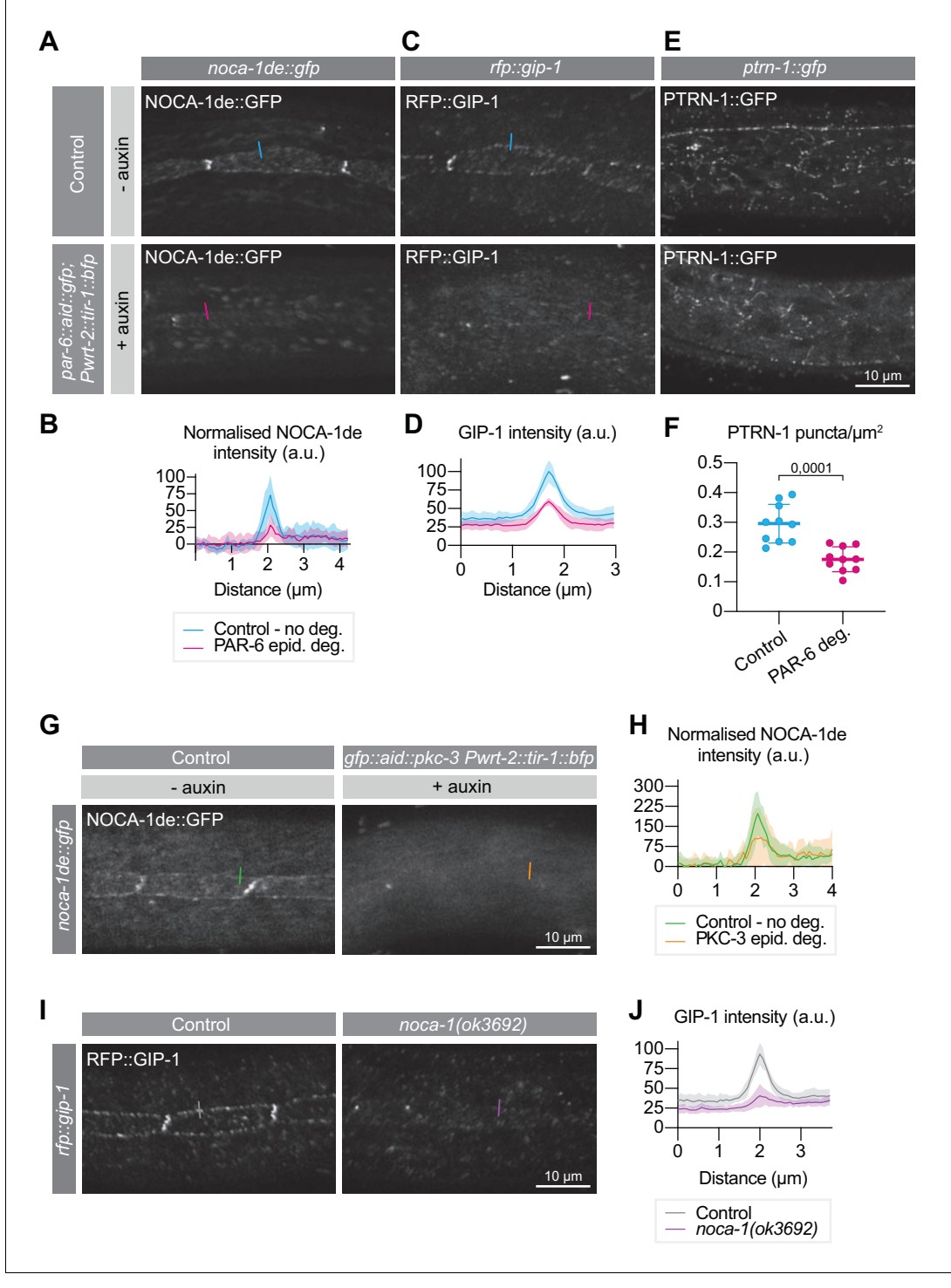

**Figure 6.** PAR-6 promotes the localization of its binding partner NOCA-1, as well as of GIP-1 and PTRN-1. (A, B) Distribution and quantification of NOCA-1de::GFP in the epidermis of *noca-1de::gfp* animals without auxin, and *noca-1de::gfp; par-6::aid::gfp; Pwrt-2::tir-1::bfp* animals in the presence of 4 mM auxin for 6 hr. n = 9 animals for Control, and 10 for PAR-6 epid. deg. Short colored lines in A indicate the area quantified in B. (C, D) Distribution and quantification of GIP-1::RFP in the epidermis of *gip-1::rfp* animals without auxin, and *gip-1::rfp; par-6::aid::gfp; Pwrt-2::tir-1::bfp* animals in the presence of 4 mM auxin for 6 hr. n = 6 for Control and 6 for PAR-6 epid. deg. Short colored lines in C indicate the area quantified in D. (E, F) Distribution and quantification of PTRN-1::GFP in the hyp7 and seam cells of *ptrn-1::gfp* animals without auxin, and *ptrn-1::gfp; par-6::aid::gfp; Pwrt-2::tir-1::bfp* animals in the presence of 4 mM auxin for 6 hr. n = 10 for Control and 10 for PAR-6 deg. Short colored lines in E indicate

*Figure 6 continued on next page*

*Figure 6 continued*

the area quantified in F. (**G, H**) NOCA-1de::GFP in the epidermis of *noca-1de::gfp* animals without auxin, and *noca-1de::gfp; gfp::aid::pkc-3; Pwrt-2::tir-1::bfp* animals in the presence of 4 mM auxin for 6 hr. n = 10 for Control and 10 for PKC-3 epid. deg. Short colored lines in G indicate the area quantified in H. (**I, J**) Distribution and quantification of GIP-1::RFP in the epidermis of *gip-1::rfp* animals and *gip-1::rfp; noca-1(ok3692)*. n = 6 for Control and 6 for *noca-1(ok3692)*. Short colored lines in I indicate the area quantified in J. All images are maximum intensity projections of the apical domain. Quantifications in B, D, H and J show mean apical GFP fluorescence intensity ± SD at the hyp7-seam-cell junction (indicated by colored lines), normalized to background intensity of each animal measured in the hypodermis. Quantification in F shows mean PTRN-1::GFP puncta density ± SD. Tests of significance: unpaired t-test for F.

The online version of this article includes the following source data and figure supplement(s) for figure 6:

**Source data 1.** Source data for *Figure 6*.
**Figure supplement 1.** Interaction of PAR-6 and NOCA-1 in the yeast two-hybrid system.
**Figure supplement 1—source data 1.** Source data for *Figure 6—figure supplement 1*.

---

alone (*Figure 5C–F*). These data are consistent with a model in which the microtubule defects caused by PAR-6 depletion are a result of the requirement of PAR-6 in localizing NOCA-1. The effects on PTRN-1 may be a secondary consequence of microtubule defects caused by NOCA-1 loss.

## Discussion

Par6 and aPKC are essential for apical–basal polarization across animal species. Most studies of the apical PAR proteins in *C. elegans* have focused on embryonic tissues, and their roles during postembryonic development remain unclear. Here, we used inducible protein degradation to identify essential roles for PAR-6 and PKC-3 in larval development. The depletion of PAR-6 or PKC-3 caused several developmental defects. When depleted from hatching, the first abnormality we observed is a severe growth defect, with animals barely increasing in length beyond their size at hatching. Surprisingly, the growth arrest is not the result of a complete developmental arrest, as the L1-stage seam-cell divisions take place at their normal time of development and the excretory canals still elongate. The next developmental defect to become apparent was the failure to complete the L1/L2 molt, indicated by incompletely released cuticles and the lack of expression of the molting marker *Pmlt-10::gfp::pest*. Both the growth defect and molting defect were rescued by expressing a non-degradable copy of PAR-6 in the hypodermis, demonstrating that PAR-6 plays an essential role in this cell type required for organismal growth and molting. Determining the cause or causes underlying the molting and growth defects will require further study. Following the L1/L2 molt we observed severe defects in the normally stereotypical pattern of seam-cell divisions, including a long delay before the next round of cell divisions. However, these defects are likely in large part a secondary consequence of the growth and molting defect, as expression of PAR-6 in the hypodermis was sufficient to restore the timing of divisions, and inducing growth and molting defects by degradation of NEKL-2 similarly induced seam-cell division defects.

In addition to these developmental defects we found that PAR-6 regulates the assembly of microtubule bundles through its interaction partner NOCA-1/Ninein. Already within 1 hr of inducing PAR-6 degradation, we observed reduced numbers of growing microtubules. This makes it unlikely that the microtubule defects are a secondary consequence of the growth or molting defects. Vice versa, a *noca-1* mutant displaying very similar microtubule defects does not display the developmental defects observed upon PAR-6 depletion. Hence, the microtubule defects are most likely independent of the growth and molting defects. The finding that depletion of PAR-6 or PKC-3 causes multiple defects likely reflects the versatility of the PAR polarity in coordinating polarity with other cellular pathways.

An essential role in postembryonic development for PAR-6 or PKC-3 has not been described. Depletion of *par-6, pkc-3,* or *par-3* by RNAi in larval stages caused defects in polarization of spermathecal cells and in ovulation, but not in larval development (*Aono et al., 2004*). Similar results were recently observed using a temperature sensitive *pkc-3* allele grown at non-permissive temperature (*Montoyo-Rosario et al., 2020*). More severe phenotypes were observed in hatching progeny

(escapers) of *par-6*, *pkc-3*, or *par-3* RNAi-treated mothers, which showed partially penetrant defects in outgrowth of vulval precursor and seam cells, migrations of neuroblasts and axons, and the development of the somatic gonad (*Welchman et al., 2007*). The lack of a growth arrest phenotype in these studies presumably reflects incomplete gene inactivation.

## Auxin-inducible protein depletion of PAR proteins

The auxin-inducible degradation approach allowed us to bypass embryonic requirements and examine the roles of PAR-6, PKC-3, and PAR-3 in specific epithelial tissues during larval development. Despite these advantages, one drawback of all protein degradation approaches is that it remains difficult to draw conclusions from negative results. Although we tagged all known PAR-3 protein isoforms and observed efficient protein depletion, ubiquitous depletion of PAR-3 did not cause obvious defects in larvae. Thus, PAR-3 may not be essential in larval tissues, and the L1 lethality previously observed for the *par-3(tm2010)* null allele (*Achilleos et al., 2010*; *Li et al., 2010a*) may be the consequence of defects in embryonic development. In support of this interpretation, PAR-3 was recently found to be largely dispensable for lumen extension of the excretory canals, in contrast to PAR-6 and PKC-3 (*Abrams and Nance, 2020*). Nevertheless, it is possible that very low levels of PAR-3 are sufficient for its functioning, or that unpredicted splicing events cause the expression of non-degron tagged PAR-3 isoforms. One approach to counteract the latter possibility would be to replace the endogenous gene with a re-engineered copy that is unlikely to express alternative splice variants, for example by replacing natural introns with artificial ones and removing internal promoters. However, removing this level of regulation and expressing only one isoform may affect the functioning of *par-3* and cause unintended side effects. We also did not detect phenotypes upon depletion of PAR-6 or PKC-3 in the larval intestine. Similar caveats as for PAR-3 depletion apply here, though PAR-6 depletion did lead to complete loss of PKC-3 from the apical domain, and epidermal depletion caused severe phenotypes. These observations make it less likely that the lack of a phenotype is due to the expression of unknown isoforms. PAR-6 and PKC-3 are likely to play nonessential or redundant roles in the intestine, as a previous study found that PAR-6 contributes to endosome positioning in this tissue (*Winter et al., 2012*).

Another advantage of inducible protein degradation is that the time it takes for defects to appear can give information on the role of the targeted protein in a particular process or structure. Processes highly dependent on the degraded protein likely show defects sooner after auxin addition than processes in which low levels of the protein suffice. Similarly, the speed at which molecular assemblies display defects will depend on whether the targeted protein is a core component of the assembly, or only required for its initial formation. For example, upon depletion of PAR-6, we observed defects in microtubule growth within 1 hr using a plus-end-binding marker, while defects in circumferential microtubule bundles visualized with a microtubule-binding protein took ~24 hr to become apparent. This indicates that PAR-6 regulates the formation of new microtubules but is not essential for the maintenance of already existing microtubule bundles. Similarly, junctional defects in the epidermis appeared ~24 after PAR-6 or PKC-3 depletion started, indicating that PAR-6 and PKC-3 are important for the assembly of new junctions, but are not integral components.

## Roles of PAR-6 and PKC-3 in junction formation and cell polarity

Depletion of PAR-6 and PKC-3 in the epidermis resulted in a fragmented appearance of the hyp7–seam and seam–seam junctions (*Figure 4E,F*), similar to previous observations in embryonic epithelia (*Montoyo-Rosario et al., 2020*; *Totong et al., 2007*; *Von Stetina et al., 2017*; *Von Stetina and Mango, 2015*). Localization of PAR-6 and PKC-3 were mutually dependent in both the epidermis and intestine. This result was not surprising, as Par6 and aPKC act as a dimer and have been shown to be mutually dependent in other *C. elegans* tissues (*Hung and Kemphues, 1999*; *Li et al., 2010b*; *Nance et al., 2003*; *Tabuse et al., 1998*; *Totong et al., 2007*). We also examined if PKC-3 functions to exclude the basolateral polarity proteins PAR-1 and LGL-1 from the apical domain. In the epidermis, PKC-3 depletion caused a rapid invasion of LGL-1 in the apical domain of the seam cells, while PAR-1 remained junctional and basal. Thus PKC-3 functions to exclude LGL-1 in the seam cells. A recent study found that LGL-1 can suppress sterility of a temperature sensitive *pkc-3* allele, further

indicating that the interaction between LGL-1 and PKC-3 is functionally relevant (*Montoyo-Rosario et al., 2020*).

In contrast to the epidermis, LGL-1 localization in the intestine remained unchanged upon PKC-3 depletion, and we observed no obvious abnormalities in the intestine upon PAR-6 or PKC-3 depletion. Thus, while PAR-6 and PKC-3 are essential for development of the embryonic intestine (*Totong et al., 2007*), they do not appear to be essential in the larval intestine. Other cellular systems, such as polarized protein trafficking, may suffice to maintain cell polarity in the absence of the apical PAR proteins (*Shafaq-Zadah et al., 2012*; *Zhang et al., 2012*; *Zhang et al., 2011*). An analogous situation exists in the *Drosophila* midgut, where integrins, but not the apical PAR proteins, are essential for polarization (*Chen et al., 2018*). The lack of LGL-1 mislocalization also points to the existence of possible redundancies in polarization of cortical polarity regulators, which may be uncovered through enhancer screens in PAR-6 or PKC-3 depleted backgrounds.

In embryonic epithelia, the requirements of the apical PAR proteins also vary between tissues. Intestinal and epidermal cells depleted of PAR-6 or PKC-3 using the ZF1 system still show apicobasal polarization, as evidenced by apical localization of junctional and cytoskeletal proteins (*Montoyo-Rosario et al., 2020*; *Totong et al., 2007*). However, in the arcade cells of the pharynx, most PAR-6 depleted animals show no apical enrichment of junctional or apical cytoskeletal markers (*Von Stetina and Mango, 2015*). These data further highlight that the requirements for PAR-6 and PKC-3 can vary between tissues.

## A novel role for PAR-6 in epidermal microtubule organization

Epidermal-specific depletion uncovered a novel role for PAR-6 in organizing non-centrosomal microtubule bundles. In epithelial cells, apical ncMTOCs assemble apical–basal microtubule arrays. ncMTOCs contain proteins and complexes involved in microtubule anchoring, microtubule stabilization, and microtubule nucleation — such as the γ-tubulin ring complex (γ-TuRC) (*Sanchez and Feldman, 2017*). How apical ncMTOCs are organized is not well understood, but several studies indicate an important role for apical PAR proteins in this process. In the cellularizing *Drosophila* embryo, aPKC is required for the transition from centrosome emanated asters to non-centrosome associated apical–basal bundles (*Harris and Peifer, 2007*). In the developing embryonic intestine of *C. elegans*, PAR-3 is needed for the redistribution of γ-tubulin and other microtubule regulators from the centrosomes to the apical domain of the cell (*Feldman and Priess, 2012*). A role for Par6 in regulating microtubule-organizing centers may not be limited to epithelial ncMTOCs. For example, in several mammalian cultured cell lines Par6 is a component of centrosomes and regulates centrosomal protein composition (*Dormoy et al., 2013*; *Kodani et al., 2010*).

Epidermal depletion of PAR-6 resulted in reduced numbers of circumferential microtubule bundles, as well as a reduced microtubule growth rate and EB1 comet density. Moreover, depletion of PAR-6 led to a loss of apical NOCA-1 enrichment at seam–seam and seam–hyp7 junctions. The effects of PAR-6 depletion on microtubule organization and dynamics are very similar to those we observed in a *noca-1* mutant, and their severity did not increase when combining PAR-6 depletion with the *noca-1* mutant. While other models are possible, these data are consistent with PAR-6 acting through NOCA-1 to control microtubule organization in the epidermis. The reduced microtubule growth rate and EB1 comet density we observed in *noca-1* mutant animals have been reported previously (*Wang et al., 2015*). However, no defects in circumferential microtubule bundle density were observed in that study, despite using the same *noca-1(ok3692)* allele. The observed difference may be a result of a difference in exact experimental procedure or the precise genetic background used. For example, whereas we used the microtubule-binding protein GFP::MAPH-1.1 to label microtubules, the study by Wang et al. used a GFP::β-tubulin fusion. However, it is not immediately clear how the markers would differentially affect microtubule density in *noca-1(ok3692)* animals, as both appear to label all microtubules, and in control animals we observe a similar microtubule density as observed by Wang et al.

We also found that, in the epidermis, the localization of GIP-1 is dependent on NOCA-1. The relationship between NOCA-1 and γ-TuRC components has been examined previously in two different tissues (*Sallee et al., 2018*; *Wang et al., 2015*, p. 1). In the germline, NOCA-1 co-localizes with γ-tubulin to non-centrosomal microtubule arrays but is not required for the localization of γ-tubulin (*Wang et al., 2015*). In fact, in this tissue the localization of a short NOCA-1 protein lacking isoform-

specific N-terminal extensions is dependent for its localization on γ-tubulin. The longer NOCA-1h isoform, however, localizes independently of γ-tubulin, indicating the presence of multiple NOCA-1 localization signals (*Wang et al., 2015*). In the embryonic intestine, the localization of NOCA-1 was not altered by the depletion of GIP-1 (*Sallee et al., 2018*). However, microtubule organization in the intestine is regulated differently from the epidermis, as apical microtubule organization was largely normal even in *ptrn-1* mutant animals depleted of intestinal NOCA-1 and GIP-1 (*Sallee et al., 2018*). Thus, differential effects of γ-TuRC component loss may reflect differences in the mechanisms of microtubule regulation. Whether PAR-6 plays a role in ncMTOC assembly and microtubule organization in tissues other than the epidermis remains to be investigated.

In addition to the effects on NOCA-1 and GIP-1, PAR-6 depletion resulted in a reduced number of PTRN-1 puncta in the epidermis. PTRN-1 is a member of the Patronin/CAMSAP/Nezha family of minus end-associated proteins, which stabilize and protect uncapped microtubule minus ends (*Atherton et al., 2019*; *Goodwin and Vale, 2010*; *Hendershott and Vale, 2014*; *Jiang et al., 2014*). NOCA-1 was previously shown to act in parallel with PTRN-1 in organizing circumferential microtubule arrays in the *C. elegans* epidermis (*Wang et al., 2015*). The mechanistic details of the relationship between NOCA-1 and PTRN-1 have not been resolved, but their distinct localization patterns suggest that they act on distinct pools of microtubules. Our data does not reveal why PAR-6 depletion results in a reduced number of PTRN-1 foci.

## Mechanisms of larval growth arrest and molting defects

The depletion of PAR-6 or PKC-3 in the epidermis led to a rapid growth arrest and failure to molt. What causes these dramatic effects? The junctional defects we observed are unlikely to be the primary consequence of either the growth or molting defects, as effects on cell junctions appeared only after 24 hr of exposure to auxin. The effects on LGL-1 were more rapid but are also not likely to explain the defects, as *lgl-1* mutants are viable (*Beatty et al., 2010*; *Hoege et al., 2010*, p. 1). The effects on the microtubule cytoskeleton are likely to contribute to the growth arrest and molting defects. However, *noca-1* mutants displayed similar microtubule defects as PAR-6 depletion yet develop to adulthood. Interestingly, *noca-1; ptrn-1* double mutant animals do grow slowly and frequently die before reaching adulthood (*Wang et al., 2015*). Thus, the combined defects in NOCA-1 and PTRN-1 localization we observed upon PAR-6 depletion may partially explain the growth defects. The roles of PTRN-1 may not be limited to microtubule regulation, as a recent study demonstrated that PTRN-1 stimulates actin polymerization during endocytic recycling in the intestine (*Gong et al., 2018*, p. 1).

Wheather and how the growth and molting defects are related is difficult to establish. The molting defect may contribute to the growth arrest, as failure to molt can cause a growth arrest (*Brooks et al., 2003*; *Lažetić and Fay, 2017*; *Russel et al., 2011*; *Yochem et al., 1999*). PAR-6 and PKC-3 could affect molting through intracellular trafficking. Molting requires the coordinated activity of the endocytic and exocytic machineries (*Lažetić and Fay, 2017*), and several links between cortical polarity regulators and the polarized trafficking machinery have been uncovered (*Rodriguez-Boulan and Macara, 2014*). In *C. elegans*, *par-3*, *par-6*, and *pkc-3* were all found to be required for endocytic trafficking in oocytes, and RNAi for *par-3* and *par-6* causes scattering of multiple endosome types in the intestine (*Balklava et al., 2007*; *Winter et al., 2012*). It is possible, therefore, that PAR-6 and PKC-3 regulate vesicle trafficking in molting as well. Such regulation may be indirect, through regulation of cytoskeletal components, or through more direct mechanisms remaining to be uncovered. However, the fact that animals in which PAR-6 or PKC-3 is depleted from hatching lack any cell growth, rather than arresting at the L1 molt, suggests that the molting defect is not the sole cause of the growth defect. Alternatively, the growth defect may contribute to the molting defect. Diet restricted animals that grow very slowly delay the L1–L2 molt until a certain body size is reached, suggesting that molting is subject to a size threshold (*Uppaluri and Brangwynne, 2015*).

In summary, our data supports that PAR-6 and PKC-3 have multiple roles in the epidermis that support larval development and molting. We have also uncovered an important role for PAR-6 in regulating the microtubule cytoskeleton, while additional mechanisms through which PAR-6 and PKC-3 control growth and/or molting likely remain to be discovered.

# Materials and methods

## Key resources table

| Reagent type (species) or resource | Designation | Source or reference | Identifiers | Additional information |
|---|---|---|---|---|
| Strain, strain background(*C. elegans*) | BOX289 | This paper | | *par-6(mib30[par-6::aid::egfp-loxp]) I; ieSi57[eft-3p::TIR1::mRuby::unc-54 3'UTR + Cbr-unc-119(+)] II* |
| Strain, strain background(*C. elegans*) | BOX570 | This paper | | *pkc-3(mib78[egfp-loxp::aid::pkc-3]) II; ieSi57[eft-3p::TIR1::mRuby::unc-54 3'UTR + Cbr-unc-119(+)] II* |
| Strain, strain background(*C. elegans*) | BOX292 | This paper | | *ieSi57[eft-3p::TIR1::mRuby::unc-54 3'UTR + Cbr-unc-119(+)] II; par-3(mib68 [eGFP-Lox2272::AID::par-3b+eGFP (noIntrons)-LoxP::AID::par-3g]) III* |
| Strain, strain background(*C. elegans*) | BOX276 | This paper | | *par-3b(mib65[eGFP-Lox2272:: AID::par-3b]) III* |
| Strain, strain background(*C. elegans*) | BOX667 | This paper | | *par-3(mib68[eGFP-Lox2272::AID::par-3b +eGFP(noIntrons)-LoxP::AID::par-3g]) III; ieSi64 [gld-1p::TIR1::mRuby::gld-1 3'UTR + Cbr-unc-119(+)] II* |
| Strain, strain background(*C. elegans*) | BOX409 | This paper | | *par-6(mib30[par-6::aid::egfp-loxp]) I; mibIs49[Pwrt-2::TIR-1::tagBFP2-Lox511:: tbb-2–3'UTR, IV:5014740–5014802 (cxTi10882 site)] IV* |
| Strain, strain background(*C. elegans*) | BOX607 | This paper | | *pkc-3(mib78[egfp-loxp::aid::pkc-3]) II; mibIs49[Pwrt-2::TIR-1::tagBFP2-Lox511::tbb-2–3'UTR, IV:5014740 –5014802 (cxTi10882 site)] IV* |
| Strain, strain background(*C. elegans*) | BOX444 | This paper | | *pkc-3(mib78[egfp-loxp::aid::pkc-3]) II; mibIs48[Pelt-2::TIR-1::tagBFP2-Lox511::tbb-2–3'UTR, IV:5014740–5014802 (cxTi10882 site)] IV* |
| Strain, strain background(*C. elegans*) | BOX285 | This paper | | *par-6(mib30[par-6::aid::egfp-loxp]) I; mibIs48[Pelt-2::TIR-1::tagBFP2-Lox511::tbb-2–3'UTR, IV:5014740 –5014802 (cxTi10882 site)] IV* |
| Strain, strain background(*C. elegans*) | BOX506 | This paper | | *par-6(mib30[par-6::aid::egfp-loxp]) I; mibIs49[Pwrt-2::TIR-1::tagBFP2-Lox511::tbb-2–3'UTR, IV:5014740 –5014802 (cxTi10882 site)] IV; dlg-1 (mib23[dlg-1::mCherry-LoxP]) X; mgIs49[mlt-10::gfp-pest]* |
| Strain, strain background(*C. elegans*) | BOX412 | This paper | | *par-6(mib30[par-6::aid::egfp-loxp]) I; mibIs49[Pwrt-2::TIR-1::tagBFP2-Lox511::tbb-2–3'UTR, IV:5014740 –5014802 (cxTi10882 site)] IV; heIs63 [Pwrt-2::GFP::PH, Pwrt-2::GFP::H2B, Plin-48::mCherry]V* |
| Strain, strain background(*C. elegans*) | BOX490 | This paper | | *ouIs10[Pscm::NLS::tdTomato(pAW584) +Pwrt2::GFP::PH(pAW561)+Pdpy-7::2x NLS::YFP(pAW516)] I; par-6(mib30[par-6 ::aid::egfp-loxp]) I; mibIs49[Pwrt-2::TIR-1 ::tagBFP2-Lox511::tbb-2–3'UTR, IV:5014740 –5014802 (cxTi10882 site)] IV* |
| Strain, strain background(*C. elegans*) | BOX041 | This paper | | *mibIs23 [lgl-1::GFP-2TEV-Avi 10 ng + Pmyo-3::mCherry 10 ng + lambda DNA 60 ng] V* |

*Continued on next page*

*Continued*

| Reagent type (species) or resource | Designation | Source or reference | Identifiers | Additional information |
|---|---|---|---|---|
| Strain, strain background(*C. elegans*) | BOX553 | This paper | | *pkc-3(mib78[egfp-loxp::aid::pkc-3]) II; mibIs49[Pwrt-2::TIR-1::tagBFP2-Lox511::tbb-2–3'UTR, IV:5014740–5014802 (cxTi10882 site)] IV; mibIs23 [lgl-1::GFP-2TEV-Avi 10 ng + Pmyo-3::mCherry 10 ng + lambda DNA 60 ng] V* |
| Strain, strain background(*C. elegans*) | BOX554 | This paper | | *pkc-3(mib78[egfp-loxp::aid::pkc-3]) II; mibIs49[Pwrt-2::TIR-1::tagBFP2-Lox511::tbb-2–3'UTR, IV:5014740–5014802 (cxTi10882 site)] IV; par-1 (it324[par-1::gfp::par-1 exon 11a])* |
| Strain, strain background(*C. elegans*) | BOX493 | This paper | | *pkc-3(mib78[egfp-loxp::aid::pkc-3]) II; mibIs49[Pwrt-2::TIR-1::tagBFP2-Lox511::tbb-2–3'UTR, IV:5014740–5014802 (cxTi10882 site)] IV; dlg-1(mib23 [dlg-1::mCherry-LoxP]) X* |
| Strain, strain background(*C. elegans*) | BOX402 | This paper | | *par-6(mib30[par-6::aid::egfp-loxp]) I; mibIs49[Pwrt-2::TIR-1::tagBFP2-Lox511::tbb-2–3'UTR, IV:5014740–5014802 (cxTi10882 site)] IV; dlg-1 (mib23[dlg-1::mCherry-LoxP]) X* |
| Strain, strain background(*C. elegans*) | BOX494 | This paper | | *mcIs40 [Plin-26::ABDvab-10::mCherry + Pmyo-2::GFP]; par-6(mib30[par-6::aid::egfp-loxp]) I; mibIs49[Pwrt-2::TIR-1::tagBFP2-Lox511::tbb-2–3'UTR, IV:5014740–5014802 (cxTi10882 site)] IV; heIs63[Pwrt-2::GFP::PH, Pwrt-2::GFP::H2B, Plin-48::mCherry] V* |
| Strain, strain background(*C. elegans*) | BOX483 | This paper | | *par-6(mib30[par-6::aid::egfp-loxp]) I; maph-1.1(mib12[egfp::maph-1.1]) I; mibIs49[Pwrt-2::TIR-1::tagBFP2-Lox511::tbb-2–3'UTR, IV:5014740–5014802 (cxTi10882 site)] IV; dlg-1(mib23[dlg-1::mCherry-LoxP]) X* |
| Strain, strain background(*C. elegans*) | BOX505 | This paper | | *maph-1.1(mib12[egfp::maph-1.1]) I; pkc-3(mib78[egfp-loxp::aid::pkc-3]) II; mibIs49[Pwrt-2::TIR-1::tagBFP2-Lox511::tbb-2–3'UTR, IV:5014740–5014802 (cxTi10882 site)] IV; dlg-1(mib23 [dlg-1::mCherry-LoxP]) X* |
| Strain, strain background(*C. elegans*) | BOX592 | This paper | | *maph-1.1(mib12[egfp::maph-1.1]) I; noca-1(ok3692)V/nT1[qIs51](IV;V)* |
| Strain, strain background(*C. elegans*) | BOX658 | This paper | | *maph-1.1(mib12)I; par-6 (mib24[par-6::egfp-loxp] I; mibIs49 [Pwrt-2::TIR-1::tagBFP2-Lox511::tbb-2–3'UTR, IV:5014740–5014802 (cxTi10882 site)]) IV; noca-1(ok3692) V/nT1[qIs51](IV;V)* |
| Strain, strain background(*C. elegans*) | BOX487 | This paper | | *par-6(mib25[par-6::mCherry-LoxP]) I; ebp-2(he293[ebp-2::egfp]) II; mibIs49[Pwrt-2::TIR-1::tagBFP2-Lox511::tbb-2–3'UTR, IV:5014740–5014802 (cxTi10882 site)] IV* |
| Strain, strain background(*C. elegans*) | BOX580 | This paper | | *ebp-2(he293[ebp-2::egfp]) II; noca-1(ok3692)V/nT1[qIs51](IV;V)* |
| Strain, strain background(*C. elegans*) | BOX659 | This paper | | *par-6(mib24[par-6::egfp-loxp] I; ebp-2(he293[ebp-2::egfp]) II; mibIs49[Pwrt-2::TIR-1::tagBFP2-Lox511::tbb-2–3'UTR, IV:5014740–5014802 (cxTi10882 site)]) IV; noca-1(ok3692)V/nT1[qIs51](IV;V)* |

*Continued*

| Reagent type (species) or resource | Designation | Source or reference | Identifiers | Additional information |
|---|---|---|---|---|
| Strain, strain background(*C. elegans*) | BOX567 | This paper | | *par-6(mib30[par-6::aid::egfp-loxp]) I; ltSi540[pOD1343/pSW160; Pnoca-1 de::noca-1de::sfGFP; cb-unc-119(+)]II; unc-119(ed3)III; mibIs49[Pwrt-2::TIR-1 ::tagBFP2-Lox511::tbb-2–3'UTR, IV: 5014740–5014802 (cxTi10882 site)] IV* |
| Strain, strain background(*C. elegans*) | BOX355 | This paper | | *par-6(mib30[par-6::aid::egfp-loxp]) I; ltSi540[pOD1343/pSW160; Pnoca -1de::noca-1de::sfGFP; cb-unc-119 (+)]II; unc-119(ed3)III; ieSi57[eft-3p:: TIR1::mRuby::unc-54 3'UTR + Cbr-unc-119(+)] II* |
| Strain, strain background(*C. elegans*) | BOX568 | This paper | | *par-6(mib30[par-6::aid::egfp-loxp]) I; gip-1(wow25[tagRFP-t::3xMyc::gip-1]) III; mibIs49[Pwrt-2::TIR-1::tagBFP2-Lox511::tbb-2–3'UTR, IV:5014740 –5014802 (cxTi10882 site)] IV* |
| Strain, strain background(*C. elegans*) | BOX502 | This paper | | *par-6(mib30[par-6::aid::egfp-loxp]) I; mibIs49[Pwrt-2::TIR-1::tagBFP2-Lox511::tbb-2–3'UTR, IV:5014740 –5014802 (cxTi10882 site)] IV; dlg-1 (mib23[dlg-1::mCherry-LoxP]) X; ptrn-1(wow4[PTRN-1::GFP]) X* |
| Strain, strain background(*C. elegans*) | BOX657 | This paper | | *pkc-3(mib78[egfp-loxp::aid::pkc-3]) II; Pnoca-1de::noca-1de::superfolder GFP; cb-unc-119(+)II; unc-119(ed3)III; mibIs49[Pwrt-2::TIR-1::tagBFP2-Lox511 ::tbb-2–3'UTR, IV:5014740–5014802 (cxTi10882 site)] IV* |
| Strain, strain background(*C. elegans*) | BOX579 | This paper | | *gip-1(wow25[tagRFP-t::3xMyc::gip-1]) III; noca-1(ok3692)V/nT1[qIs51](IV;V)* |
| Strain, strain background(*C. elegans*) | BOX561 | This paper | | *par-6(mib30[par-6::aid::egfp-loxp]) I; mibIs49[Pwrt-2::TIR-1::tagBFP2-Lox511::tbb-2–3'UTR, IV:5014740 –5014802 (cxTi10882 site)] IV; mibEx221(Pdpy-7::par-6::mch)* |
| Strain, strain background(*C. elegans*) | BOX563 | This paper | | *par-6(mib30[par-6::aid::egfp-loxp]) I; mibIs49[Pwrt-2::TIR-1::tagBFP2-Lox511::tbb-2–3'UTR, IV:5014740–5014802 (cxTi10882 site)] IV; heIs63[Pwrt-2::GFP:: PH, Pwrt-2::GFP::H2B, Plin-48::mCherry] V; mibEx222(Pdpy-7::par-6::mch; Pmyo-2::egfp)* |
| Strain, strain background(*C. elegans*) | BOX608 | This paper | | *pw27[nekl-2::aid];pwSi10 [phyp7::bfp::tir-1];pw17[chc-1::GFP]; mibEx223(Pwrt-2::mCh::H2B; Pwrt-2::mCh::PH)* |
| Strain, strain background(*C. elegans*) | BOX447 | This paper | | *pkc-3(mib78[egfp-loxp::aid::pkc-3]) II; mibIs48[Pelt-2::TIR-1::tagBFP2-Lox511::tbb-2–3'UTR, IV:5014740 –5014802 (cxTi10882 site)] IV; mibIs23 [lgl-1::GFP-2TEV-Avi 10 ng + Pmyo-3::mCherry 10 ng + lambda DNA 60 ng] V* |
| Strain, strain background(*C. elegans*) | BOX431 | This paper | | *par-6(mib30[par-6::aid::egfp-loxp]) I; mibIs48[Pelt-2::TIR-1::tagBFP2-Lox511::tbb-2–3'UTR, IV:5014740 –5014802 (cxTi10882 site)] IV; dlg-1 (mib23[dlg-1::mCherry-LoxP]) X* |

*Continued*

| Reagent type (species) or resource | Designation | Source or reference | Identifiers | Additional information |
|---|---|---|---|---|
| Strain, strain background(*C. elegans*) | BOX406 | This paper | | *par-6(mib30[par-6::aid::egfp-loxp]) I; pkc-3(mib80[mcherry-loxp::pkc-3]) II; mibIs48[Pelt-2::TIR-1::tagBFP2-Lox511 ::tbb-2–3'UTR, IV:5014740–5014802 (cxTi10882 site)] IV* |
| Strain, strain background(*C. elegans*) | BOX653 | This paper | | *par-6(mib24[par-6::egfp-loxp] I; pkc-3(mib78[egfp-loxp::aid::pkc-3]) II; mibIs48[Pelt-2::TIR-1::tagBFP2-Lox511::tbb-2–3'UTR, IV:5014740–5014802 (cxTi10882 site)]) IV* |
| Strain, strain background(*C. elegans*) | BOX411 | This paper | | *par-6(mib30[par-6::aid::egfp-loxp]) I; pkc-3(mib80[mcherry-loxp::pkc-3]) II; mibIs49[Pwrt-2::TIR-1::tagBFP2-Lox511::tbb-2–3'UTR, IV:5014740–5014802 (cxTi10882 site)] IV* |
| Strain, strain background(*C. elegans*) | BOX578 | This paper | | *par-6(mib30[par-6::aid::egfp-loxp]) I; par-3(it300[par-3::mcherry]) III; mibIs49[Pwrt-2::TIR-1::tagBFP2-Lox511::tbb-2–3'UTR, IV:5014740–5014802 (cxTi10882 site)] IV* |
| Strain, strain background(*C. elegans*) | BOX484 | This paper | | *par-6(mib25[par-6::mCherry-LoxP]) I; pkc-3(mib78[egfp-loxp::aid::pkc-3]) II; mibIs49[Pwrt-2::TIR-1::tagBFP2-Lox511 ::tbb-2–3'UTR, IV:5014740–5014802 (cxTi10882 site)] IV* |
| Strain, strain background(*C. elegans*) | BOX485 | This paper | | *pkc-3(mib78[egfp-loxp::aid::pkc-3]) II; par-3(it300[par-3::mcherry]) III; mibIs49 [Pwrt-2::TIR-1::tagBFP2-Lox511::tbb-2–3' UTR, IV:5014740–5014802 (cxTi10882 site)] IV* |
| Strain, strain background(*C. elegans*) | BOX486 | This paper | | *par-6(mib25[par-6::mCherry-LoxP]) I; par-3(mib68[eGFP-Lox2272::AID:: par-3b+eGFP(noIntrons)-LoxP::AID:: par-3g]) III; mibIs49[Pwrt-2::TIR-1:: tagBFP2-Lox511::tbb-2–3'UTR, IV: 5014740–5014802 (cxTi10882 site)] IV* |
| Strain, strain background(*C. elegans*) | BOX492 | This paper | | *pkc-3(it309[GFP::pkc-3]) II; par-3(mib68[eGFP-Lox2272::AID ::par-3b+eGFP(noIntrons)-LoxP::AID ::par-3g]) III; mibIs49[Pwrt-2::TIR-1:: tagBFP2-Lox511::tbb-2–3'UTR, IV: 5014740–5014802 (cxTi10882 site)] IV* |
| Strain, strain background(*C. elegans*) | AW1015 | *Hughes et al., 2014* | RRID:WB-STRAIN: WBStrain00042230 | *ouIs10[Pscm::NLS::tdTomato (pAW584)+Pwrt2::GFP::PH (pAW561)+Pdpy-7::2xNLS::YFP (pAW516)] I* |
| Strain, strain background(*C. elegans*) | BOX188 | *Waaijers et al., 2016* | | *maph-1.1(mib12[egfp::maph-1.1]) I* |
| Strain, strain background(*C. elegans*) | CA1200 | CGC | RRID:WB-STRAIN: WBStrain00004055 | *ieSi57[eft-3p::TIR1::mRuby:: unc-54 3'UTR + Cbr-unc-119(+) ] II; unc-119(ed3) III* |
| Strain, strain background(*C. elegans*) | GR1395 | *Hayes et al., 2006* | RRID:WB-STRAIN: WBStrain00007913 | *mgIs49 [mlt-10::GFP-pest; ttx-1::GFP]* |
| Strain, strain background(*C. elegans*) | JLF15 | Jessica Feldman | | *ptrn-1(wow4[PTRN-1::GFP]) X* |
| Strain, strain background(*C. elegans*) | JLF173 | Jessica Feldman | | *gip-1(wow25[tagRFP-t:: 3xMyc::gip-1]) III* |
| Strain, strain background(*C. elegans*) | KK1218 | CGC | RRID:WB-STRAIN: WBStrain00023582 | *par-3(it300[par-3::mCherry]) III* |

*Continued on next page*

*Continued*

| Reagent type (species) or resource | Designation | Source or reference | Identifiers | Additional information |
|---|---|---|---|---|
| Strain, strain background(*C. elegans*) | KK1228 | CGC | RRID:WB-STRAIN: WBStrain00023583 | *pkc-3(it309[GFP::pkc-3]) II* |
| Strain, strain background(*C. elegans*) | KK1262 | CGC | RRID:WB-STRAIN: WBStrain00023586 | *par-1 (it324[par-1::gfp::par-1 exon 11a])* |
| Strain, strain background(*C. elegans*) | ML916 | CGC | RRID:WB-STRAIN: WBStrain00026581 | *mcIs40 [Plin-26::ABDvab-10:: mCherry + Pmyo-2::GFP]* |
| Strain, strain background(*C. elegans*) | OD1652 | Karen Oegema | RRID:WB-STRAIN: WBStrain00044359 | *ltSi540[pOD1343/pSW160; Pnoca-1de::noca-1de::sfGFP; cb-unc-119(+)]II; unc-119(ed3)III* |
| Strain, strain background(*C. elegans*) | RT3638 | David Fay | | *pw27[nekl-2::aid];pwSi10 [phyp7::bfp::tir-1];pw17[chc-1::GFP]* |
| Strain, strain background(*C. elegans*) | SV1009 | *Wildwater et al., 2011* | RRID:WB-STRAIN: WBStrain00034608 | *heIs63[Pwrt-2::GFP::PH, Pwrt-2::GFP::H2B, Plin-48::mCherry]V* |
| Strain, strain background(*C. elegans*) | SV1937 | Sander van den Heuvel | | *ebp-2(he293[ebp-2::egfp]) II* |
| Strain, strain background(*C. elegans*) | VC2998 | CGC | RRID:WB-STRAIN: WBStrain00037614 | *noca-1(ok3692)V/nT1[qIs51](IV;V)* |
| Strain, strain background(*C. elegans*) | CA1352 | | RRID:WB-STRAIN: WBStrain00004071 | *ieSi64 [gld-1p::TIR1::mRuby::gld-1 3'UTR + Cbr-unc-119(+)] II* |
| Strain, strain background(*C. elegans*) | STR320 | Martin Harterink | | *maph-1.1(mib15[GFPKI]);hrtEx110 [Pptrn-1::ebp-2::mKate2; Pmyo-2::tdTom]* |
| Recombinant DNA reagent | Plasmid: pJJR82 | Addgene | #75027 | EGFPSEC3xFlag vector with ccdB markers for cloning homology arms |
| Recombinant DNA reagent | Plasmid: pJJR83 | Addgene | #75028 | mCherrySEC3xFlag vector with ccdB markers for cloning homology arms |
| Recombinant DNA reagent | Plasmid: pMLS257 | Addgene | #73716 | SapTrap destination vector for building repair template only vectors |
| Recombinant DNA reagent | Plasmid: pDD379 | Addgene | #91834 | SapTrap destination vector for building combined sgRNA expression + repair template vectors, using the F+E sgRNA scaffold |
| Recombinant DNA reagent | Plasmid: pJJR50 | Addgene | #75026 | U6 promoter driven flipped + extended sgRNA expression vector |
| Recombinant DNA reagent | Plasmid: Peft-3::cas9 | Addgene | #46168 | codon optimized Cas9_SV40 NLS with intron |
| Recombinant DNA reagent | Plasmid: Pdpy-7::par-6::mCherry | This paper | | Plasmid for expression of PAR-6:: mCherry in the hypodermis (*Figure 3—figure supplement 2*). Full sequence in *Supplementary file 1*. |
| Recombinant DNA reagent | Plasmid: PAR-6 PDZ in pMB28 | This paper | | Yeast expression plasmid of PAR-6 PDZ fused to Gal4 DNA binding domain (*Figure 6—figure supplement 1*). Full sequence in *Supplementary file 1*. |
| Recombinant DNA reagent | Plasmid: NOCA-1d in pMB29 | This paper | | Yeast expression plasmid of NOCA-1d fused to Gal4 activation domain (*Figure 6—figure supplement 1*). Full sequence in *Supplementary file 1*. |
| Sequence-based reagent | par-6 sgRNA | gacgcaaatgacagtgatagTGG | | sgRNA target site used to engineer the *par-6* locus. PAM site in uppercase. |
| Sequence-based reagent | pkc-3 sgRNA | tgggtctccgacatcattagAGG | | sgRNA target site used to engineer the *pkc-3* locus. PAM site in uppercase. |

*Continued on next page*

*Continued*

| Reagent type (species) or resource | Designation | Source or reference | Identifiers | Additional information |
|---|---|---|---|---|
| Sequence-based reagent | par-3 sgRNA 1 | tttcagatcgatcatcatgtCGG | | sgRNA target site used to target the *par-3* locus. PAM site in uppercase. |
| Sequence-based reagent | par-3 sgRNA 2 | cacatgcataacggtcgtggTGG | | sgRNA target site used to target the *par-3* locus. PAM site in uppercase. |
| Sequence-based reagent | dlg-1 sgRNA | gccacgtcattagatgaaatTGG | | sgRNA target site used to target the *dlg-1* locus. PAM site in uppercase. |
| Sequence-based reagent | mos IV 5013690. 5015700 sgRNA | agctcaatcgtgtacttgcgTGG | | sgRNA target site for LG IV position 5013690.5015700, used to insert TIR-1 expression cassette. PAM site in uppercase. |
| Sequence-based reagent | ebp-2 sgRNA 1 | gcaggcaaatctggacgataCGG | | sgRNA target site used to edit the *ebp-2* locus. |
| Sequence-based reagent | ebp-2 sgRNA 2 | tacggggataggataagcaaTGG | | sgRNA target site used to edit the *ebp-2* locus. |
| Sequence-based reagent | Pdpy-7_F | This paper | PCR primers | TGTAATACGACTCACTATAGGGCGAA TTGGctcattccacgatttctcgc. See Materials and methods section 'PAR-6::mCherry transgenic array' for usage details. |
| Sequence-based reagent | Pdpy-7_R | This paper | PCR primers | tctggaacaaaatgtaagaatattc See Materials and methods section 'PAR-6::mCherry transgenic array' for usage details. |
| Sequence-based reagent | par-6_F1 | This paper | PCR primers | tttaagaatattcttacattttgttccagaATGT CCTACAACGGCTCCTA See Materials and methods section 'PAR-6::mCherry transgenic array' for usage details. |
| Sequence-based reagent | par-6_R1 | This paper | PCR primers | GGCCATGTTGTCCTCCTCTCCCTTGGAC ATGTCCTCTCCACTGTCCGAAT See Materials and methods section 'PAR-6::mCherry transgenic array' for usage details. |
| Sequence-based reagent | par-6_UTR_F | This paper | PCR primers | CACTCCACCGGAGGAATGGACGAGCTC TACTGAaaaactctttcagcca See Materials and methods section 'PAR-6::mCherry transgenic array' for usage details. |
| Sequence-based reagent | par-6_UTR_R | This paper | PCR primers | TAAAGGGAACAAAAGCTGGAGCTCCACCGC gaaataaataatttattctc See Materials and methods section 'PAR-6::mCherry transgenic array' for usage details. |
| Sequence-based reagent | mCherry_F | This paper | PCR primers | TCCAAGGGAGAGGAGGACAA See Materials and methods section 'PAR-6::mCherry transgenic array' for usage details. |
| Sequence-based reagent | mCherry_R | This paper | PCR primers | GTAGAGCTCGTCCATTCCTC See Materials and methods section 'PAR-6::mCherry transgenic array' for usage details. |
| Sequence-based reagent | par-6_F2 | This paper | PCR primers | ggaggcgcgccATGATTGTGCCAGAAGCTCATCG See Materials and methods section 'yeast two-hybrid' analysis for usage details. |

*Continued on next page*

*Continued*

| Reagent type (species) or resource | Designation | Source or reference | Identifiers | Additional information |
|---|---|---|---|---|
| Sequence-based reagent | par-6_R2 | This paper | PCR primers | ggagcggccgcTCAGGCGTTCGGTGTTCCTTGTT See Materials and methods section 'yeast two-hybrid' analysis for usage details. |
| Sequence-based reagent | noca-1d_F | This paper | PCR primers | ggaggcgcgccATGAATATTTGTTGTTGTGG See Materials and methods section 'yeast two-hybrid' analysis for usage details. |
| Sequence-based reagent | noca-1d_R | This paper | PCR primers | ggagcggccgcCTATTGAACTCTGCATACAT. See Materials and methods section 'yeast two-hybrid' analysis for usage details. |

## *C. elegans* strains

All *C. elegans* strains used in this study are derived from the N2 Bristol strain, and are listed in the Key resources table. All strains were maintained at 20°C on Nematode Growth Medium (NGM) plates seeded with *Escherichiae coli* OP50 bacteria under standard conditions (*Brenner, 1974*).

## CRISPR/Cas9 genome engineering

All gene editing was done by homology-directed repair of CRISPR/Cas9-induced DNA double-strand breaks, using plasmid-based expression of Cas9 and sgRNAs. All edits were made in an N2 background, with the exception of 2x(*egfp::aid)::par-3*, for which *egfp::aid::par-3* was used as the starting background. All fusions were repaired using a plasmid-based template with 190–600 bp homology arms and containing a self-excising cassette (SEC) for selection (*Dickinson et al., 2015*). The homology arms included mutations of the sgRNA recognition sites to prevent re-cutting after repair. The *par-6::aid::egfp*, *par-6::mCherry*, *dlg-1::mCherry* and *ebp-2::egfp* vectors were cloned using Gibson assembly and vector pJJR82 (Addgene #75027) (*Gibson et al., 2009*; *Ramalho et al., 2020*) as the backbone. The *2x(egfp::aid)::par-3*, *Pwrt-2::tir-1::bfp* and *Pelt-2::tir-1::bfp* vectors were cloned using SapTrap assembly into vector pMLS257 (Addgene #73716) (*Schwartz and Jorgensen, 2016*), and the *egfp::aid::pkc-3* and *mCherry::pkc-3* vectors were cloned using SapTrap assembly into vector pDD379 (Addgene #91834) (*Dickinson et al., 2018*). The sgRNAs were expressed from a plasmid under control of a U6 promoter. To generate sgRNA vectors, antisense oligonucleotide pairs were annealed and ligated into BbsI-linearized pJJR50 (Addgene #75026) (*Waaijers et al., 2016*), with the exception of the *pkc-3* fusions, in which the sgRNA was incorporated into assembly vector pDD379 using SapTrap assembly. The targeted sequences can be found in Table 2. Injection mixes were prepared in MilliQ $H_2O$ and contained 50 ng/μl *Peft-3::cas9* (Addgene ID #46168) (*Friedland et al., 2013*), 50–100 ng/μl *U6::sgRNA*, 50 ng/μl of repair template, with the exception of the *pkc-3* fusions, in which the sgRNA-repair-template vector was used at a concentration of 65 ng/μl. All mixes also contained 2.5 ng/μl of the co-injection pharyngeal marker *Pmyo-2::GFP* or *Pmyo-2::tdTomato* to aid in visual selection of transgenic strains. Young adult hermaphrodites were injected in the germline using an inverted micro-injection setup (Eppendorf FemtoJet 4x mounted on a Zeiss Axio Observer A.1 equipped with an Eppendorf Transferman 4 r). Candidate edited progeny were selected on plates containing 250 ng/μl of hygromycin (*Dickinson et al., 2015*), and correct genome editing was confirmed by Sanger sequencing (Macrogen Europe) of PCR amplicons encompassing the edited genomic region. From correctly edited strains, the hygromycin selection cassette was excised by a heat shock of L1 larvae at 34°C for 1 hr in a water bath. Correct excision was confirmed by Sanger sequencing. Sequence files of the final gene fusions in Genbank format are in *Supplementary file 1*.

## *C. elegans* synchronization

In order to obtain synchronized worm populations, plates with eggs were carefully washed with M9 (0.22 M $KH_2PO_4$, 0.42 M $Na_2HPO_4$, 0.85 M NaCl, 0.001 M $MgSO_4$) buffer in order to remove larvae

and adults but leave the eggs behind. Plates were washed again using M9 buffer after 1 hr, to collect larvae hatched within that time span.

## Auxin-inducible degradation

Auxin treatment was performed by placing synchronized populations of worms on NGP plates seeded with *E. coli* OP50 and containing 1 or 4 mM auxin. To prepare plates, auxin (Alfa Aesar A10556) was diluted into the autoclaved NGM agar medium after cooling to 60°C prior to plate pouring. Plates were kept for a maximum of 2 weeks in the dark at 4°C.

## *C. elegans* growth curves

To measure growth curves, L1 animals synchronized as described above were placed on NGM plates seeded with *E. coli* OP50 and either lacking auxin or containing 4 mM auxin. Images were taken in 24 hr intervals up to 96 hr, using a Zeiss Axio Zoom.V16 equipped with a PlanNeoFluar Z 1x/0.25 objective and Axiocam 506 color camera, driven by Zen Pro software. Animal length was quantified in ImageJ(FIJI) software by drawing a spline along the center line of the animal (*Rueden et al., 2017*; *Schindelin et al., 2012*).

## Molting assay

Synchronized L1 animals were placed on NGM plates seeded with *E. coli* OP50 and either lacking auxin or containing 1 mM auxin. Fluorescence images were taken in 1 hr intervals from 11 hr to 32 hr of development, using a Zeiss Axio Zoom.V16 equipped with a PlanNeoFluar Z 1x/0.25 objective and Axiocam 506 color camera, driven by Zen Pro software. Expression levels of the *Pmlt-10::gfp::pest* reporter were quantified in ImageJ(FIJI) software (see image analysis).

## Microscopy

Live imaging of *C. elegans* larvae was done by mounting larvae on 5% agarose pads in a 10 mM Tetramisole solution in M9 buffer to induce paralysis. DIC imaging was performed with an upright Zeiss AxioImager Z2 microscope using a $63 \times 1.4$ NA objective and a Zeiss AxioCam 503 monochrome camera, driven by Zeiss Zen software. Spinning disk confocal imaging was performed using a Nikon Ti-U microscope driven by MetaMorph Microscopy Automation and Image Analysis Software (Molecular Devices) and equipped with a Yokogawa CSU-X1-M1 confocal head and an Andor iXon DU-885 camera, using $60 \times$ or $100 \times 1.4$ NA objectives. All stacks along the z-axis were obtained at 0.25 µm intervals, and all images were analyzed and processed using ImageJ(FIJI) and Adobe Photoshop. For quantifications, the same laser power and exposure times were used within experiments.

## Quantitative image analysis

All image analysis was done in using ImageJ (FIJI). For intensity profile measurements of spinning disk microscopy data, background values were subtracted from the intensity measurements. Mean background intensity was quantified on a circular region in an area not containing any animals, except in quantifications in *Figure 4A*; *Figure 4C*; *Figure 6A*; *Figure 6G*, where background intensity was quantified on a circular region in an area with no fluorescence inside the worm.

For the intensity profiles in the epidermis, except those of RFP::GIP-1, a 10 px-wide line was drawn in the apical focal plane, from the hyp7 cytoplasm to the seam-cell cytoplasm. The position of the line was chosen to avoid fluorescent signals present in neighboring tissues, notably the intestine and excretory canal. Additionally, mCherry tagged proteins tend to aggregate, as is evident from comparison with the otherwise identical GFP tagged variants. Hence, mCherry intensity profile lines were positioned to avoid aggregates. The RFP::GIP-1 fusion proteins localize in a punctate pattern. To accurately capture the average intensity of this marker protein, we drew 10 separate 20-px wide lines per cell, which covers 25–50% of the total seam-cell circumference. Intensity values were manually aligned at their peak values, and then averaged to obtain a single intensity profile per cell. For the intensity profiles in the intestine, we drew 8 separate 50 px-wide lines from the intestinal lumen to the cytoplasm of the intestinal cells, which were aligned at their peak values and averaged to obtain a single value per worm. The intensity profiles from multiple animals were manually aligned at the peak values for analysis and display.

To quantify the fluorescence intensity for the molting assay, whole worm fluorescence was quantified. A region of interest (ROI) of each whole worm was created by drawing a freehand line around the worm using the transmitted light channel. The corresponding fluorescence of the ROI was measured in the GFP channel.

Microtubule bundles were counted manually as follows: a 5-px-wide freehand line was drawn through an ~80 µm stretch of microtubule bundles at the dorsal or ventral region of an animal, and the intensity profile was plotted. The number of fluorescent peaks was counted, and the microtubule bundle density was calculated by dividing the number of peaks by the measured distance.

EBP-2::GFP comet counting was done manually as follows: an ROI was drawn around the area of hyp7 visible in the camera field of view (corresponding to 300–500 µm$^2$). The entire width of hyp7 was included, from the outside of the animal up to (but excluding) the seam cells. Either the ventral or dorsal hypodermis was analyzed. Comets were counted manually within the ROI, and density was calculated by dividing the number of EBP-2::GFP comets by the surface of the area analyzed.

PTRN-1::GFP puncta counting was done manually. The entire epidermal area visible in the camera field of view was analyzed, and puncta in both the seam cells and the hypodermis were counted. Puncta density was calculated by dividing the number of PTRN-1::GFP puncta by the surface of the area analyzed.

Microtubule growth rate was calculated in an automated manner using the ImageJ plug-in 'TrackMate' (*Tinevez et al., 2017*). An ROI was drawn around either the seam cells or hyp7 visible in the camera field of view. For hyp7 either the ventral or dorsal area was analyzed. The following parameters were chosen: estimated blob diameter = 0.700 um; threshold = 200,000; simple LAP tracker; linking max distance = 1.5 um; gap-closing max distance = 1.5 um; gap-closing max frame gap = 3; duration of track = 10. The mean speed of the comets was averaged to obtain the average microtubule growth rate. Comets in both the seam cells and the hypodermis were measured and represented separately.

To determine the directionality of the actin bundles and microtubule growth, images or movies were rotated to orient the seam cells horizontally. Lines were drawn along the microtubule or actin bundles, and the angle of these lines was calculated relative to the horizontal axis. Per animal, an area containing 20 actin bundles or 30 microtubule bundles was analyzed (all bundles in the area were analyzed). Movies of EBP-2 were used to calculate the directionality of microtubule growth, where the direction of growth of individual comets was annotated manually. Maximum intensity projections of EBP-2 movies were used to calculate the directionality of microtubule growth. Rose plots were generated using MatLAB.

## Relative excretory canal outgrowth

To quantify relative canal outgrowth in the excretory canal cell, synchronized animals were placed on NGM plates seeded with *E. coli* OP50 and either lacking auxin or containing 4 mM auxin. Animals were placed on plates immediately after hatching. The distance between the cell body and either the anterior distal body tip or the anus was determined by tracing a segmented line along the center of the animal. The length of each individual canal was measured with a segmented line from the anterior-posterior bifurcation points close to the cell body until the canal tip. Relative outgrowth was calculated as the fraction of canal length over the distance between the cell body and the anterior distal tip or the anus.

## Seam lineage analysis

To generate the seam-cell lineage, synchronized animals were placed on NGM plates seeded with *E. coli* OP50 and either lacking auxin or containing 1 mM auxin. Animals were placed on plates immediately after hatching (before L1 degradation), at 7 hr of development (before L2 degradation) or at 19 hr of development (before L3 degradation). At 1 hr intervals, 5–10 animals were randomly picked and transferred to a microcopy slide. The number of seam cells and hyp7 nuclei were determined manually based on expression of the dual-color marker *ouIs10[scmp::NLS::tdTomato; dpy-7p::2xNLS::YFP;wrt-2p::GFP::PH]* that marks the seam nuclei in red and the hypodermal nuclei in green. Divisions of V5 were excluded from the analysis as V5 follows a different division pattern at the L2 stage, in which the anterior daughter becomes a neuroblast that generates a sensory structure termed the posterior deirid sensillium. V5 cells are readily recognized based on their position in

the row of the seam cells, and in L2 stage additionally on their division pattern. Animals were classified according to showing a wild-type seam-cell division pattern, having developmental defects such as delayed or arrested seam-cell divisions, or having inappropriate seam-cell differentiation. Control animals were classified at each larval stage. PAR-6 depleted animals were classified after they had undergone the delayed L2-stage divisions. From the total number of worms analyzed, the percentages of worms in each category were calculated.

### PAR-6::mCherry transgenic array

The *Pdpy-7::par-6::mCherry* plasmid used for PAR-6 hypodermal rescue was cloned into the pBSK(+) vector using Gibson assembly. The promoter of *dpy7*, which is expressed in hyp7 but not in the seam cells (*Gilleard et al., 1997*; *Myers and Greenwald, 2005*), was amplified from *C. elegans* genomic DNA using primers Pdpy-7_F and Pdpy-7_R. A fragment of 5.3 kb containing the entire genomic sequence of *par-6* and a fragment of 402 bp of the *par-6* 3' UTR were amplified from *C. elegans* N2 genomic DNA using primers par-6_F1 and par-6_R1, and par-6_UTR_F and par-6_UTR_R, respectively. mCherry was amplified from pJJR83 (Addgene #75028) using primers mCherry_F and mCherry_R. Correct amplification and assembly were confirmed by Sanger sequencing. The plasmid generated can be found in *Supplementary file 1*. See Key resources table for primer sequences. To generate transgenic lines young adult hermaphrodites were injected in the germline with 30 ng/µl of *Pdpy-7::par-6::mCherry*. mCherry fluorescence was used to select stable transgenic lines.

### Yeast two-hybrid analysis

Sequences encoding the PAR-6 PDZ domain and full-length NOCA-1d were PCR amplified from a mixed-stage cDNA library using primers par-6_F2 and par-6_R2, and noca-1d_F and noca-1d_R. See Key resources table for primer sequences. PCR products were digested with AscI and NotI, and cloned into Gal4-DB vector pMB28 and Gal4-AD vector pMB29, respectively (*Koorman et al., 2016*). The resulting plasmids were transformed into *Saccharomyces cerevisiae* strains Y8930 (MATα) and Y8800 (MATa) (*Yu et al., 2008*) using the Te/LiAc transformation method (*Schiestl and Gietz, 1989*). DB::PAR-6/AD::NOCA-1 diploid yeast was generated by mating, and plated on synthetic defined (SD) medium plates lacking leucine, tryptophan, and histidine containing 2 mM 3-Amino-1,2,4-triazole (3-AT); and lacking leucine, tryptophan, and adenine to assess the presence of an interaction, and on an SD plate lacking leucine and histidine containing 1 µg/ml cycloheximide to test for self-activation by the DB::PAR-6 plasmid in the absence of the AD::NOCA-1 plasmid. Controls of known reporter activation strength and behavior on cycloheximide were also added to all plates.

### Statistical analysis

All statistical analyses were performed using GraphPad Prism 8. For population comparisons, a D'Agostino and Pearson test of normality was first performed to determine if the data was sampled from a Gaussian distribution. For data drawn from a Gaussian distribution, comparisons between two populations were done using an unpaired t-test, with Welch's correction if the SDs of the populations differ significantly, and comparisons between >2 populations were done using a one-way ANOVA, or a Welch's ANOVA if the SDs of the populations differ significantly. For data not drawn from a Gaussian distribution, a non-parametric test was used (Mann-Whitney for two populations and Kruskal-Wallis for >2 populations). ANOVA and non-parametric tests were followed up with multiple comparison tests of significance (Dunnett's, Tukey's, Dunnett's T3 or Dunn's). Tests of significance used and sample sizes are indicated in the figure legends. No statistical method was used to pre-determine sample sizes. No samples or animals were excluded from analysis. The experiments were not randomized, and the investigators were not blinded to allocation during experiments and outcome assessment.

## Acknowledgements

We thank R Schmidt and S van den Heuvel for sharing strain SV1937, S van den Heuvel for strain SV1009, J Feldman for strains JLF15 and JLF173, K Oegema for strain OD1652, A Woollard for strain AW1015, A Frand for strain GR1395, and D Fay for strain RT3638. We thank S van den Heuvel, M Harterink, D Fay and members of the S van den Heuvel and M Boxem groups for helpful discussions,

M Harterink for critical reading of the manuscript, J Sepers for help generating PAR-3 strains, and J Cravo for generating the rose plots. We also thank Wormbase (*Harris et al., 2020*) and the Biology Imaging Center, Faculty of Sciences, Department of Biology, Utrecht University. Some strains were provided by the Caenorhabditis Genetics Center, which is funded by NIH Office of Research Infrastructure Programs (P40 OD010440). This work was supported by the Netherlands Organization for Scientific Research (NWO)-ALW Open Program 824.14.021 and NWO-VICI 016.VICI.170.165 grants to M Boxem, and the European Union's Horizon 2020 research and innovation programme under the Marie Skłodowska-Curie grant agreement No. 675407 – PolarNet.

## Additional information

### Funding

| Funder | Grant reference number | Author |
|---|---|---|
| Nederlandse Organisatie voor Wetenschappelijk Onderzoek | 824.14.021 | Mike Boxem |
| Nederlandse Organisatie voor Wetenschappelijk Onderzoek | 016.VICI.170.165 | Mike Boxem |
| H2020 Marie Skłodowska-Curie Actions | ITN 675407 | Mike Boxem |

The funders had no role in study design, data collection and interpretation, or the decision to submit the work for publication.

### Author contributions

Victoria G Castiglioni, Helena R Pires, Conceptualization, Formal analysis, Investigation, Visualization, Methodology, Writing - original draft, Writing - review and editing; Rodrigo Rosas Bertolini, Formal analysis, Investigation; Amalia Riga, Jana Kerver, Investigation; Mike Boxem, Conceptualization, Supervision, Funding acquisition, Visualization, Writing - original draft, Project administration, Writing - review and editing

### Author ORCIDs

Victoria G Castiglioni ⓘ https://orcid.org/0000-0001-9884-2537
Helena R Pires ⓘ https://orcid.org/0000-0001-8351-0812
Mike Boxem ⓘ https://orcid.org/0000-0003-3966-4173

### Decision letter and Author response

Decision letter https://doi.org/10.7554/eLife.62067.sa1
Author response https://doi.org/10.7554/eLife.62067.sa2

## Additional files

### Supplementary files

• Supplementary file 1. A zip archive containing the DNA sequences of genome edits and plasmids described in this paper in genbank and SnapGene format.

• Transparent reporting form

### Data availability

All quantative data generated during this study are included in the manuscript and supporting files. All microscopy images shown are supplied as source data files.

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
