## [Decision Letter]

Thank you for submitting your article "Epidermal PAR-6 and PKC-3 are essential for larval development of *C. elegans* and organize non-centrosomal microtubules" for consideration by *eLife*. Your article has been reviewed by three peer reviewers, and the evaluation has been overseen by a Reviewing Editor and Suzanne Pfeffer as the Senior Editor. The reviewers have opted to remain anonymous.

The reviewers have discussed the reviews with one another and the Reviewing Editor has drafted this decision to help you prepare a revised submission.

The editors have judged that your description of postembryonic roles of PAR proteins are of interest. However, additional experiments are required before the manuscript can be accepted for publication. As you will see described in detail below, a number of issues were raised about the way that some of the phenotypes were scored. In addition, genetic support for the PAR/Noca interaction is a critical, missing piece of information. Together with a number of other points, these requests should be straight-forward to address and all reviewers agree on their importance. Note that because of the Covid-19 situation, we do not require that revised manuscripts are returned within 2 months.

Reviewer #1:

In this study, Castiglioni and colleagues use the AID system to examine the roles of Par proteins Par3, Par6 and aPKC during larval growth. They find that epidermal expression of PAR-6 and PKC-3 (but not PAR-3 as tagged) is required for larval development and molting, and for the pattern of seam-cell divisions. Epistasis experiments using the tagged alleles support some previously published findings, but also suggest possible complexities that were not previously appreciated (such as regulation of PAR-3 localization by PKC-3 but not PAR-6 in seam cells). The most novel aspect of the paper is the finding that PAR-6 is important for non-centrosomal MT regulation, and may do so through its binding partner NOCA-1, although this aspect of the paper would benefit from additional experimentation.

Overall, the data presented are rigorous and I agree with most interpretations, save a few below that are surprising and would benefit from additional experimental tests. Much of the paper reveals findings that are not unexpected based on previous studies in the embryo or other model systems. The most novel and exciting aspect of the paper is the connection between PAR-6 and non-centrosomal MTs, which the authors propose occurs through NOCA-1 recruitment (which the Boxem lab revealed as a PAR-6-binding protein in a previous study). This conclusion should be strengthened by additional genetic tests, such as simultaneous removal of PAR-6 and NOCA-1 to compare phenotypes to the single mutants. In addition, the impact of the paper to the field would be increased if it were shown how PAR-6 interfaces with NOCA-1. Is it simply recruiting NOCA-1 to the apical surface? Does PKC-3 phosphorylation contribute as well, or is the role of PKC-3 in this context limited to localizing PAR-6?

Reviewer #2:

In this manuscript, Castiglioni and colleagues analyzed the role of PAR proteins during larval development in *C. elegans*. While the function of PARs in embryonic development is well established, their role during larval development is poorly characterized. The authors developed an auxin-inducible depletion system of PAR proteins and showed that PAR-6 and PKC-3 are required in the epidermis for larval development, molting, proper polarity and cell division patterns. They also observed that PAR-6 regulates the pattern of microtubules via its direct interaction with NOCA-1/Ninein. Overall, this is a high quality study. The conclusions are well supported by the experiments and the manuscript is nicely written. The novel interaction between PAR-6 and NOCA-1 is of broad interest and can justify publication in *eLife*.

There is one point of interest that the authors did not really discuss. In the hyp7 epidermal cell, microtubules are oriented circumferentially (perpendicular to the anteroposterior axis). They observed that PAR-6 depletion perturbed this orientation and the direction of microtubule growth (Figure 5C, 5H). This suggests that, in addition to its role in apicobasal polarity and junction organization, PAR-6 could also have a role in the planar polarity of epidermal cells. Could that be due to PAR-6 enrichment at the hyp7-seam borders ? It would be interesting to discuss how PAR-6 could instruct the microtubule orientation.

Reviewer #3:

This paper describes requirements of PAR proteins in *C. elegans* larval epithelium using degron system. It shows that PAR-6 and PKC-3 in epidermis are necessary for growth of animals, molting, junctional protein localizations and organizations of non-centrosomal microtubules. However, the manuscript contains a number of problems. Substantial revision and additional experiments are necessary to understand the function of par proteins in epidermis.

1) PAR-6 or PKC-3 depletion causes various defects, however, relationship among them are not clear and difficult to understand, especially since different time points after the auxin treatment were used for each analysis: 24, 48 and 72hrs for growth, 5 and 12hrs for junction, 6 and 24hrs for junctional protein localizations, 1hr for PAR protein localizations, 24hrs for MT density and 1hr for ebp-1 signals. Phenotypes at the same time point (e.g. 6hrs) should be shown in addition to other time points and thoroughly describe and discuss order of their appearance after the treatments.

2) For the growth defects, according to the graphs and raw data, sizes of the depleted animals after 24hrs are similar to those at 1-2.5 hrs hatching (very strangely, except for Figure 3—figure supplement 1B that contradicts with Figure 2I) which indicate that animals stop growing just after the treatment, raising the possibility that most other defects are secondary consequences of the former. In addition, molting defect is likely to affect phenotypes beyond the L2 stage (e.g. 24hrs). It is surprising that animals start molting and undergo L2 stage seam-cell divisions without any growth. Although Figure 3E shows L2 seam divisions, it clearly contradicts with no increase in seam-cell numbers shown in Figure 3—figure supplement 1D. Therefore, I wonder if the lineaged animals (Figure 3E) and animals with molting defects (Figure 3A) are rare escaper from the growth defects. Most of the depleted animals may just arrest at the L1 stage. Since dpy-7::PAR-6 can rescue growth defects but not seam-cell morphology, lineage should be analyzed in this rescued animals to know functions of PAR-6 in seam cells.

3) In the figure legends of many images used for quantification (Figure 2 AB, Figure 4AC, Figure 4—figure supplement 2A-F, Figure 6A), no explanation about the colored lines, "distance" in the graphs and how the quantification was done. Part of such information can only be found in Materials and methods section. But no explanation how positions of the lines are determined. I guess the authors intentionally decide positions of lines (at least this possibility cannot be excluded). Some of the images (Figure 4—figure supplement 2A-D) are quite dirty with background(?) signals. The authors appears to avoid those signals. But not clear how real and background signals are distinguished. They should have fair criteria about positions of the lines, e.g. above the center of seam-cell nuclei. They can avoid background signals only when there are clear reasons to do so. In addition, although it was described "The intensity profiles were manually aligned at the apical peak value" in the Materials and methods section, this is not the case at least for Figure 4B (I checked the source file only for this).

---

## [Author Response]

Reviewer #1:In this study, Castiglioni and colleagues use the AID system to examine the roles of Par proteins Par3, Par6 and aPKC during larval growth. They find that epidermal expression of PAR-6 and PKC-3 (but not PAR-3 as tagged) is required for larval development and molting, and for the pattern of seam cell divisions. Epistasis experiments using the tagged alleles support some previously published findings, but also suggest possible complexities that were not previously appreciated (such as regulation of PAR-3 localization by PKC-3 but not PAR-6 in seam cells). The most novel aspect of the paper is the finding that PAR-6 is important for non-centrosomal MT regulation, and may do so through its binding partner NOCA-1, although this aspect of the paper would benefit from additional experimentation.Overall, the data presented are rigorous and I agree with most interpretations, save a few below that are surprising and would benefit from additional experimental tests. Much of the paper reveals findings that are not unexpected based on previous studies in the embryo or other model systems. The most novel and exciting aspect of the paper is the connection between PAR-6 and non-centrosomal MTs, which the authors propose occurs through NOCA-1 recruitment (which the Boxem lab revealed as a PAR-6-binding protein in a previous study). This conclusion should be strengthened by additional genetic tests, such as simultaneous removal of PAR-6 and NOCA-1 to compare phenotypes to the single mutants.

Examining the double inactivation is a great suggestion, and one we should have thought of for the initial submission. We have now performed PAR-6 depletion in the *noca-1* mutant background and have included the results in the manuscript. We do not see an increase in the severity of microtubule disorganization (using both the MAPH-1.1 and EBP-2 markers). These results strengthen our conclusion that the microtubule disorganization upon PAR-6 depletion is due to the loss of NOCA-1 localization.

In addition, the impact of the paper to the field would be increased if it were shown how PAR-6 interfaces with NOCA-1. Is it simply recruiting NOCA-1 to the apical surface? Does PKC-3 phosphorylation contribute as well, or is the role of PKC-3 in this context limited to localizing PAR-6?

We liked this line of thought a lot and wanted to test this by inactivating the kinase function of PKC-3. In a paper from the Goehring lab (Rodrigues et al., 2017), the authors used a pkc-3 temperature sensitive allele and a drug to inactivate the kinase function of pkc-3. We thought either approach would be ideal to address this point, as neither approach disrupted membrane localization of PAR-6 and PKC-3 in the one-cell embryo. We contacted the drug manufacturer via several channels, but did not receive any replies. We also crossed the ts allele to our microtubule organization markers and to the LGL-1 marker strain. Unfortunately, at the restrictive temperature, we did not see any apical displacement of LGL-1 (in contrast to our depletion results). We also saw no effect on microtubule organization using MAPH-1.1 as the marker, but without displacement of LGL-1 this result is meaningless as we have no indication that the kinase function of PKC-3 is actually disrupted. As pkc-3(ts) mutants become sterile adults, a much milder phenotype than what we observe by protein degradation, it seems likely that the ts allele indeed does not fully disrupt PKC-3 kinase activity. We therefore were not able to address this question.

Reviewer #2:In this manuscript, Castiglioni and colleagues analyzed the role of PAR proteins during larval development in *C. elegans*. While the function of PARs in embryonic development is well established, their role during larval development is poorly characterized. The authors developed an auxin-inducible depletion system of PAR proteins and showed that PAR-6 and PKC-3 are required in the epidermis for larval development, molting, proper polarity and cell division patterns. They also observed that PAR-6 regulates the pattern of microtubules via its direct interaction with NOCA-1/Ninein. Overall, this is a high quality study. The conclusions are well supported by the experiments and the manuscript is nicely written. The novel interaction between PAR-6 and NOCA-1 is of broad interest and can justify publication in eLife.There is one point of interest that the authors did not really discuss. In the hyp7 epidermal cell, microtubules are oriented circumferentially (perpendicular to the anteroposterior axis). They observed that PAR-6 depletion perturbed this orientation and the direction of microtubule growth (Figure 5C, 5H). This suggests that, in addition to its role in apicobasal polarity and junction organization, PAR-6 could also have a role in the planar polarity of epidermal cells. Could that be due to PAR-6 enrichment at the hyp7-seam borders ? It would be interesting to discuss how PAR-6 could instruct the microtubule orientation.

We did notice the PAR-6 enrichment and the hyp7-seam borders, which is consistent with the recent observation by the Michaux lab that the apical PAR module is planar polarized in the lateral embryonic epidermis of the embryo. In the larval stages, however, the planar polarization of PAR-6 was not consistent, and whether it localized planar or more uniformly did not appear to correlate with any particular stage in the seam lineage pattern. For this reason we did not highlight it in the manuscript. Whether the PAR module has a role in planar polarity is difficult to say for sure. The seam cell defects we observed, in particular the orientation defects, do hint at a problem with determining anterior/posterior direction, but could also be a consequence of defects in another cellular process. The microtubule network in the seam cells is not regularly organized, nor visibly disrupted upon loss of PAR-6. We felt that a discussion of planar polarity would therefore be highly speculative. Similarly, we have not data to indicate that, in addition to NOCA-1 recruitment, PAR-6 directly instructs microtubule orientation in the hypodermis. We have now mentioned the planar localization of PAR-6 with a reference to the data from the Michaux lab, but have not added an in depth discussion. We hope the reviewer will agree with us that such a discussion would be too speculative.

Reviewer #3:This paper describes requirements of PAR proteins in *C. elegans* larval epithelium using degron system. It shows that PAR-6 and PKC-3 in epidermis are necessary for growth of animals, molting, junctional protein localizations and organizations of non-centrosomal microtubules. However, the manuscript contains a number of problems. Substantial revision and additional experiments are necessary to understand the function of par proteins in epidermis.1) PAR-6 or PKC-3 depletion causes various defects, however, relationship among them are not clear and difficult to understand, especially since different time points after the auxin treatment were used for each analysis: 24, 48 and 72hrs for growth, 5 and 12hrs for junction, 6 and 24hrs for junctional protein localizations, 1hr for PAR protein localizations, 24hrs for MT density and 1hr for ebp-1 signals. Phenotypes at the same time point (e.g. 6hrs) should be shown in addition to other time points and thoroughly describe and discuss order of their appearance after the treatments.

We understand that the occurrence of multiple different phenotypes upon PAR-6 or PKC-3 depletion can be confusing. Most likely, and in line with current thinking in the field, PAR-6/PKC-3 are involved in multiple processes, and it is expected that multiple phenotypes are uncovered. We have tried to clarify the potential relationships among the phenotypes by opening the discussion with a summary of the effects observed, and which ones we think are secondary and which are independent. We also extensively rewrote the section “Cell autonomous and non-autonomous roles for PAR-6 and PKC-3 in molting, seam cell divisions and seam cell morphology”. We emphasize that the growth defect does not correspond to a general developmental arrest, and added new data showing that excretory canal outgrowth during L1 continues. We also highlight the possible relationships between growth and molting, as molting defects block growth, but molting itself has also been reported to require growth. We already had written and discussed that the seam cell division defects largely are a secondary consequence. Finally, we tried to be careful in the manuscript text to describe the time points analyzed and times at which auxin was added. We have gone through the text and added further time point statements.

With regards to the analysis timepoints, we based these either on the time in development that a certain event occurs (e.g. seam division, the first molt, or appearance of circumferential actin bundles), or on the earliest moment the defect appears (e.g. 1 h for ebp-1 defects and 24 h for MT density defects). Picking a single time point is not possible, as either the event we want to show is not taking place at that time, or the defect is not evident yet. We feel it is important to show the earliest time point at which a defect becomes apparent, as it is in fact informative with regards to the role of the targeted protein. The speed at which a defect appears after the targeted protein is depleted gives an indication on whether the degraded protein is an integral component of the pathway or structure examined, or is required for assembly but not functioning of a structure. For example, upon depletion of PAR-6, we observed defects in microtubule growth or nucleation within 1 h, while defects in circumferential microtubule bundles took ~24 h to become apparent. This indicates that PAR-6 regulates the formation of new microtubules but is not essential for the maintenance of already existing microtubule bundles. Similarly, junctional defects in the epidermis appeared ~24 after PAR-6 or PKC-3 depletion started, indicating that PAR-6 and PKC-3 are important for the assembly of new junctions, but are not integral components. We have added a discussion of this topic to the Discussion section.

2) For the growth defects, according to the graphs and raw data, sizes of the depleted animals after 24hrs are similar to those at 1-2.5 hrs hatching (very strangely, except for Figure 3—figure supplement 1B that contradicts with Figure 2I) which indicate that animals stop growing just after the treatment, raising the possibility that most other defects are secondary consequences of the former. In addition, molting defect is likely to affect phenotypes beyond the L2 stage (e.g. 24hrs). It is surprising that animals start molting and undergo L2 stage seam cell divisions without any growth. Although Figure 3E shows L2 seam divisions, it clearly contradicts with no increase in seam cell numbers shown in Figure 3—figure supplement 1D. Therefore, I wonder if the lineaged animals (Figure 3E) and animals with molting defects (Figure 3A) are rare escaper from the growth defects. Most of the depleted animals may just arrest at the L1 stage. Since dpy-7::PAR-6 can rescue growth defects but not seam cell morphology, lineage should be analyzed in this rescued animals to know functions of PAR-6 in seam cells.

Below we address each of the individual points. Overall, we have rewritten this section of the Results extensively to clarify the questions of the reviewer, and clarify which aspects of the observed phenotypes we think are secondary consequences. We also elaborate on the possible relationships between growth and molting defects, in both the Results and Discussion, where we are now much more careful in describing that growth and molting are interrelated.

– The animals do indeed stop growing just after the treatment. The interpretation that Figure 3—figure supplement 1B contradicts with Figure 2I is, however, not correct. Figure 3—figure supplement 1B shows animals in which we exogenously express wild-type PAR-6 in hyp7. These animals only lack PAR-6 in the seam cells and do indeed grow. This experiment is part of our evidence that the growth defect is due to a role of PAR-6 in the hypodermis.

– Figure 3E also does not contradict with Figure 3—figure supplement 1D. In Figure 3—figure supplement 1D, we show the 24h timepoint. To clarify, we have added timing marks in hours to Figure 3E. Wild-type animals at 24h point have completed the L2 divisions that cause a doubling of cell number. PAR-6 depleted animals have such a long delay that there has been no doubling division. This is why control animals in Figure 3—figure supplement 1D show 10-20 cells, while the PAR-6 depleted animals show 5 seam cells. The third bar shows that expression of PAR-6 in hyp7 is sufficient to rescue the seam cell division timing. The reason for the spread from 10-20 cells is that after division, anterior seam cells fuse with the hypodermis. Hence immediately after division there are 20 cells, and after fusion is complete there are 10 cells again. At the 24h timepoint some cells have fused and some have not.

– We have certainly not analyzed rare escapers. For the growth assays, a number of animals are placed on a plate, and all animals are analyzed at each timepoint. For the lineaging, synchronized populations are washed off the plate, placed on a microscopy slide, and analyzed in the order they are spotted under the microscope.

– We do not think that PAR-6 depletion causes a full L1 developmental arrest, as aspects of development clearly continue despite the growth arrest. In all animals, the L1 seam cell divisions take place at the normal time. We have now also measured elongation of the excretory canals during L1 development as another marker of development. The visible molting defects in Figure 3A are not rare: 30-50% of animals show molting defects indicating that they reached the first molt but fail to complete the molt.

– We do agree that these findings are surprising: that aspects of development continue to take place despite the growth arrest. We have rewritten the section “Cell autonomous and nonautonomous roles for PAR-6 and PKC-3 in molting, seam cell divisions and seam cell morphology” to more clearly state that this is the case, and include the new excretory canal data.

– We completely agree that the growth and molting defects likely cause secondary defects. This is why we performed the experiments shown in Figure 3—figure supplement 1. They show that when we express non-degradable PAR-6 in hyp7, we not only restore the growth and molting defects, but also rescue most of the seam cell lineage defects. Moreover, arresting growth and molting by another method (nekl-2 depletion) causes a defect in seam cell divisions similar to PAR-6 depletion. We have rewritten the results to clarify that the seam cell division defects likely are a secondary defect, and added an extensive discussion of the order in which phenotypes appear and their relationships.

– Finally, while we did not perform a full lineaging analysis of dpy-7::PAR-6 rescued animals, Figure 3—figure supplement 1D shows that in these animals, L2 divisions take place as normal. Hence the severe seam cell lineage defects are due to loss of PAR-6 in hyp7, and PAR-6 is dispensable in the seam cells (at least for cell division timing). A full lineage analysis would be very time consuming and would not give additional information on the functioning of PAR-6 in the seam cells.

3) In the figure legends of many images used for quantification (Figure 2 AB, Figure 4AC, Figure 4—figure supplement 2A-F, Figure 6A), no explanation about the colored lines, "distance" in the graphs and how the quantification was done. Part of such information can only be found in Materials and methods section. But no explanation how positions of the lines are determined. I guess the authors intentionally decide positions of lines (at least this possibility cannot be excluded). Some of the images (Figure 4—figure supplement 2A-D) are quite dirty with background(?) signals. The authors appears to avoid those signals. But not clear how real and background signals are distinguished. They should have fair criteria about positions of the lines, e.g. above the center of seam cell nuclei. They can avoid background signals only when there are clear reasons to do so. In addition, although it was described "The intensity profiles were manually aligned at the apical peak value" in the Materials and methods section, this is not the case at least for Figure 4B (I checked the source file only for this).

We have expanded the Materials and methods section to include more details on quantification. Lines were drawn manually rather than at a predetermined location to avoid two types of aberrant signal that would distort the measurements. (1) areas where the cell being quantified was abutting another cell showing fluorescence. The seam cells, hypodermis, excretory canal, and intestine are all close to each other and all express polarity proteins. It is common therefore to observe fluorescence from multiple cells in one imaging plane. (2) Known background mCherry aggregation signals. As reviewer 1 mentions, mCherry is unfortunately prone to aggregation. However, because we also have the corresponding GFP fusions we know that the aggregates are indeed background. We fully agree with the reviewer that the “dirty” mCherry signals complicate analysis. We address this point in detail in our response to reviewer 1’s questions regarding Figure 4—figure supplement 2.

Most of the markers we used have a uniform localization pattern, and the measurements use a 10 px wide line to average out minor fluctuations in fluorescence (the indicative lines in the drawing are only ~1 px wide). Combined with measuring multiple animals for each experimental condition, this should yield an accurate representative value. The only marker we measured that does not have a uniform localization is GIP-1, which localizes in a punctate pattern. For this marker we drew extra wide lines (20 px) and drew 10 non-overlapping lines per cell, thus covering 25-50% of the total cell circumference.

We have added an explanation of the colored lines to all legends. We have expanded the Materials and methods with the above information on how the position of the lines was determined. We are unsure what kind of explanation the reviewer would like to see for an explanation of distance. The graphs already indicate that the distance is in micrometer, and the short, colored lines indicate where the measurements were taken.

We have double checked the source file for Figure 4B, but it is aligned properly. The values for each line measured are in the columns, and each column was shifted up or down such that the maximum values (reddest color) of all columns align horizontally. The shifting is clearly visible at the top and bottom of the sheet.